# Two Halves Make a Whole:
# How to Reconcile Soundness and Robustness in Watermarking for Large Language Models

## Abstract

Watermarking techniques have been used to safeguard AI-generated content. In this paper, we study publicly detectable watermarking schemes (Fairoze et al.) of LLM, and have several research findings.

First, we observe that two important security properties, robustness and soundness, may conflict with each other. We then formally investigate these two properties in the presence of an arguably more realistic adversary that we called editing-adversary, and we can prove an impossibility result that, the robustness and soundness properties cannot be achieved via a publicly-detectable **single** watermarking scheme. Second, we demonstrate our feasible result: we for the first time introduce the new concept of publicly-detectable **dual** watermarking scheme, for AI-generated content. We provide a novel construction by using two publicly-detectable watermarking schemes; each of the two watermarking schemes can achieve "half" of the two required properties: one can achieve robustness, and the other can achieve soundness. Eventually, we can **combine the two halves into a whole**, and achieve the robustness and soundness properties at the same time. Our construction has been implemented and evaluated based on OPT-2.7B , LLaMA-7B and Mistral.

## 1 Introduction

***Generative AI and robust watermarking.*** Generative AI technologies, especially advancements in large language models (LLMs), exhibit a broad range of impressive capabilities. However, these powerful tools also present risks, such as the potential for misuse in spreading fabricated or false information. To address these cybersecurity concerns, watermarking schemes have been proposed to safeguard AI-generated content Kirchenbauer et al. (2023); Aaronson (2023); Kuditipudi et al. (2023). These schemes embed a watermark into the output text during LLM generation, with the primary goal of ensuring that the watermark remains detectable even if the text is modified by an adversary.

***Achieving both robustness and soundness properties, using watermarking.*** Two important security properties, robustness and soundness, have been formalized Christ et al. (2023); Fairoze et al. (2023). In Christ et al. (2023), the soundness property is formally defined. To achieve the soundness property, a construction has been developed. Concretely, a "secret watermark" is embedded in the output of the generative model, by using a secret key. When a text is received, we can check whether the text has been watermarked or not by using the secret key. The downside of the above mentioned *privately detectable* watermarking mechanism is obvious: the generative model and the detector must share the same secret key, and a party is not allowed to detect LLM-generated content if he/she is not aware of the secret information that has been embedded in the content. Very recently, *publicly detectable* watermarking for AI-generated content is proposed in Fairoze et al. (2023). With this new primitive, any party is allowed to detect if a content is watermarked or not.

***A technical difficulty.*** Unfortunately, we observe that there is a technical difficulty in achieving soundness and robustness properties *at the same time*. Intuitively, the robustness property requires that even if a watermarked text has been modified, the embedded watermark should not be eliminated; instead, it should still be able to be detected. We remark that, an adversary could simply remove the entire watermarked text with the goal of eliminating the embedded watermark. To avoid this trivial

attack, in the formalization for the robustness property in Fairoze et al. (2023), the adversary is not allowed to remove the entire watermarked text; instead, the modified version from the adversary, denoted as $t'$, and the original version of the watermarked text, denoted as $t$, must have an overlapping of at least a $\delta$-length segment, where $\delta \in \mathbb{N}$. To better illustrate our ideas, here let's use $t' \bowtie_\delta t$ to denote the $\delta$-length segment overlapping between text $t'$ and text $t$. On the other hand, the soundness property requires that an adversary, after seeing multiple watermarked texts, say $t_1, t_2, \ldots, t_q$, should not be able to generate a valid (i.e., detectable) but "different" watermarked text $t'$. Here difference means there is no overlapping of a $k$-length window between two texts $t'$ and $t$, we write as $t' \not\bowtie_k t$, where $k \in \mathbb{N}$. For all texts, it is required that $(t' \not\bowtie_k t_1) \wedge (t' \not\bowtie_k t_2) \wedge \cdots \wedge (t' \not\bowtie_k t_q)$.

We must note that, the conditions in the two properties are conflicting with each other. Robustness requires that the modified text has a sufficient overlap ($\delta$-length) with the original text, while soundness requires that the generated text does not have a sufficient overlap ($k$-length) with the original text. Let $t \in \{t_i\}_{1 \leq i \leq q}$. These two properties will lead to the following dilemma.

For simplicity, Let $\ell$ be the actual length of the longest overlapping segment of $t'$ and $t$.

- Case 1 ($\delta \geq k$): If $\ell \geq \delta$, then the condition $t' \bowtie_\delta t$ is satisfied. However, since $\ell \geq k$, the condition $t' \not\bowtie_k t$ is not met. Conversely, if $\ell < \delta$, then $t' \bowtie_\delta t$ is not satisfied. Therefore, we conclude that no modified text $t'$ can simultaneously satisfy both $t' \bowtie_\delta t$ and $t' \not\bowtie_k t$ in Case 1.

- Case 2 ($\delta < k$): If $\ell < \delta$, then $t' \bowtie_\delta t$ is not satisfied. If $\ell \geq k$, then $t' \not\bowtie_k t$ is not satisfied. If $\delta \leq \ell < k$ and the robustness property holds, meaning the watermark can be detected from $t'$, then the soundness property is violated.

More concretely, **in Case 1**, if the length of the overlapping segment between $t'$ and $t$ is greater than or equal to $\delta$ (i.e., $\ell \geq \delta$), then $t'$ overlaps with $t$ by more than $k$. Consequently, $t'$ does not satisfy the assumption of the soundness property, which states that $t' \not\bowtie_k t$. Conversely, if the length of the overlapping segment is less than $\delta$ (i.e., $\ell < \delta$), then $t'$ does not meet the assumption of the robustness property, which requires $t' \bowtie_\delta t$. **In Case 2**, the length of the overlapping segment between $t'$ and $t$ can satisfy the assumptions of both the robustness and soundness properties (i.e., $\delta \leq \ell < k$). However, if the watermark can be detected from $t'$, the soundness property is violated; otherwise, the robustness property is compromised.

***Our research question.*** Based on the above discussions, we have the following question:

> *Is it possible to achieve the* robustness and soundness *properties* **at the same time***, in a publicly detectable watermarking scheme for LLM-generated content?*

## 1.1 OUR CONTRIBUTIONS

We give an affirmative answer to the above research question. In this paper, we carry out a systematic study on publicly detectable watermarking for LLM-generated content. We want to highlight that, we are the *first* to introduce the new concept of publicly detectable **dual** watermarking, for LLM-generated content. Concretely, we have the following results.

### 1.1.1 EDITING ADVERSARIES AND PUBLICLY DETECTABLE **SINGLE** WATERMARKING

**New adversaries with edit distance.** We first remark that in Fairoze et al. (2023), the differences between texts are measured using the length of overlapping substrings. This way of measuring differences is not strict enough, as an adversary could change small amounts of text at specific positions to avoid long consecutive substrings. In natural language, a more reasonable way to measure the differences of text is based on *edit distance*. Edit distance is the minimal steps that are needed to modify a text to another one. We emphasize that it is non-trivial using edit distance to describe texts embedding watermark because small edit distance cannot guarantee the integrity of the watermark. We are the *first* to consider a restricted but arguably more realistic adversary, that we called *editing-adversary*, with the goal of providing a better understanding of the security properties when we study watermarking for LLM-generated content. Here, considering the text generated by the adversary and the text generated by the generative model, if the difference is measured by edit distance, then the adversary is called an editing-adversary.

**A formal treatment for publicly detectable single watermarking.** If in a watermarking scheme, the watermark can be detected publicly, it is defined as publicly detectable watermarking in Fairoze et al. (2023). If the watermark detector returns a unique boolean value to indicate if the watermark is detected in the watermarking scheme, we observe that the robustness and soundness security properties may conflict with each other. We define this type of watermarking scheme as publicly detectable single watermarking.

**An impossibility result in the presence of editing adversaries.** We redefine soundness and robustness in the presence of editing-adversary. We now are able to formally investigate if the two conflicting properties, soundness and robustness, can be achieved at the same time or not, for a publicly detectable single watermarking. Indeed, we can formally establish an *impossibility result* for achieving soundness and robustness at the same time in the presence of an editing-adversary, if we use a single watermarking scheme.

### 1.1.2 Publicly detectable **dual** watermarking against editing adversaries

**A new concept: Publicly detectable dual watermarking.** To bypass the impossibility result, we introduce a new primitive, publicly detectable **dual** watermarking, for LLM-generated content. We formally define the syntax and the required properties, including robustness and soundness, of the new primitive. We remark that, the impossibility result of achieving robustness and soundness at the same time, does not hold for the dual watermarking scheme anymore.

**A new construction of publicly detectable dual watermarking scheme.** We then demonstrate our feasibility result by constructing a publicly detectable **dual** watermarking scheme. In our construction, we use two publicly detectable watermarking schemes as building blocks. Note that neither scheme can achieve soundness and robustness at the same time in the presence of an editing-adversary; however, the two watermarking schemes can achieve "half" of the two required properties, respectively: one can achieve robustness, and the other can achieve soundness. Interestingly, we are able to **combine the two halves into a whole**, and achieve the robustness and soundness properties at the same time! In this way, we successfully reconcile the two properties in watermarking for LLMs.

**Implementation and evaluation.** We implement our publicly detectable dual watermarking scheme based on OPT-2.7B Zhang et al. (2022) , LLaMA-7B Touvron et al. (2023) and Mistral Jiang et al. (2023). We then evaluate the probability that a watermark bit is embedded correctly; we also evaluate the quality of the text which is affected by watermark embedding. Our experiments show that, with a small tune factor the watermark can be embedded with very high probability. Our experiments further show that in our dual watermarking scheme, the text quality is reduced marginally. Finally, our experiments demonstrate that the parameter selection made in the theoretical parts of the paper is achievable.

### 1.2 Organization

The paper is organized as follows. Section 2 covers the preliminaries, including formal definitions for publicly detectable watermarking and the building blocks of our constructions. Section 3 redefines the security properties and proves the impossibility result. In Section 4, we introduce a novel definition of publicly detectable dual watermarking and its security properties. Section 5 presents our main construction, with security proofs in Appendix F. Section 6 discusses the implementation and evaluation results. A brief overview of related work is provided in Section 7, followed by the conclusion in Section 8.

Finally, in Section A, we include the related work including AI-generated content detection and watermarking schemes for LLM. We provide detailed preliminaries in Appendix B, definition of properties in Appendix C, supporting materials for analysis in Appendix D, details of publicly-detectable dual watermarking construction in Appendix E and additional experiments result in Appendix G.

## 2 Preliminaries

We use $\lambda$ to denote the security parameter. A negligible function $\mathsf{negl}(\lambda)$ are those functions that decay faster than the inverse of any polynomials in $\lambda$. In this paper, we describe each text $t$ generated

by the LLM as *a vector of tokens* $x_1, \ldots, x_n$; we write it as $\boldsymbol{t} = x_1 \| \cdots \| x_n$. We let $\epsilon$ denote the empty vector. We define the length of the text $\boldsymbol{t}$ as $|\boldsymbol{t}|$, which represents the number of tokens in the text, denoted as $|\boldsymbol{t}| = n$. We use the symbol $\boldsymbol{t}[i]$ to denote the $i$-th token $x_i$ of the token-vector $\boldsymbol{t}$. When the context is clear, we often also refer to a token as *a word*, and a vector of tokens as *a string*. We use substring $\hat{\boldsymbol{t}}$ to denote any consecutive tokens in $\boldsymbol{t}$ such as $\hat{\boldsymbol{t}} = x_i \| x_{i+1} \| \cdots \| x_j$ where $1 \le i \le j \le n$. For simplicity, we use $\boldsymbol{t}[i :]$ to denote the substring of $\boldsymbol{t}$ from the $i$-th element to the end; that is $\boldsymbol{t}[i :] = x_i \| \cdots \| x_n$. When we append a token $x$ to a vector $\boldsymbol{t}$, we write it as $\boldsymbol{t} \| x$. Finally, we use $\mathcal{V}$ to represent the *token vocabulary*; we use $\mathcal{V}^*$ to denote texts *with arbitrary lengths* where tokens are from $\mathcal{V}$.

**Building Blocks.** In our construction we uses cryptographic hash functions, digital signature scheme and error-correcting code (ECC) as building blocks. We also use edit distance to limit how a text $\boldsymbol{t}$ can be modified by adversary. Due to space limitations in the main text, we have placed the formal definitions in the Appendix B.

## 2.1 PUBLICLY-DETECTABLE WATERMARKING OF LLM

In this paper, we explore the watermark embedding algorithm in a large language model, commonly referred to as LLM. The large language model is a probabilistic generative model. We follow the definitions in Kirchenbauer et al. (2023); Christ et al. (2023); Fairoze et al. (2023), as below:

**Definition 2.1** (Auto-regressive Model). An auto-regressive model Model over vocabulary $\mathcal{V}$ takes prompt $\boldsymbol{\rho} \in \mathcal{V}^*$ and the previous output of the model $\boldsymbol{t} \in \mathcal{V}^*$ as input. Then it outputs a vector of logits of each word in the vocabulary as $\mathcal{D} \xleftarrow{\$} \mathsf{Model}(\boldsymbol{\rho}, \boldsymbol{t})$.

**Definition 2.2** (Generative Language Model). A generative language model GenModel over vocabulary $\mathcal{V}$ takes prompt $\boldsymbol{\rho} \in \mathcal{V}^*$ and generated text $\boldsymbol{t}$ as input. Then it outputs a sequence of words in $\mathcal{V}$ with length $n$.

In the generative language model (GenModel), the auto-regressive model $\mathsf{Model}(\cdot)$ serves as the foundation, with a prediction algorithm $\mathsf{Predict}(\cdot)$ utilized to choose the subsequent output token, as outlined in Algorithm 1. Most commonly, $\mathsf{Predict}(\cdot)$ normalizes the logits values of $\mathcal{D}$ and takes the token $x$ with the highest probability as the output.

---

**Algorithm 1** Generative Language Model (GenModel)

**Input:** $\boldsymbol{\rho}, \boldsymbol{t}, n$
  **for** $i = 1, \ldots, n$ **do**
    $\mathcal{D} \xleftarrow{\$} \mathsf{Model}(\boldsymbol{\rho}, \boldsymbol{t})$
    $\boldsymbol{t} \leftarrow \boldsymbol{t} \| \mathsf{Predict}(\mathcal{D})$
  **end for**
  return $\boldsymbol{t}$

---

**Syntax.** Our focus in this paper is on publicly detectable watermarking for LLM. We adopt the definition of a publicly detectable watermarking scheme (PDWS) as presented in Fairoze et al. (2023).

**Definition 2.3** (Publicly-Detectable Watermarking Scheme). A publicly detectable watermarking scheme PDWS for a generative language model GenModel over token vocabulary $\mathcal{V}$ consists of a tuple of algorithms $\mathsf{PDWS} = (\mathsf{Setup}, \mathsf{Watermark}, \mathsf{Detect})$ where:

- The setup algorithm $(\mathsf{pk}, \mathsf{sk}) \xleftarrow{\$} \mathsf{Setup}(1^\lambda)$. The algorithm Setup takes as input a security parameter $1^\lambda$ and outputs a pair of public and private keys $(\mathsf{pk}, \mathsf{sk})$.

- The watermarking algorithm $\boldsymbol{t} \xleftarrow{\$} \mathsf{Watermark}(\mathsf{sk}, \boldsymbol{\rho})$. The algorithm Watermark takes as input a private key sk and a prompt $\boldsymbol{\rho} \in \mathcal{V}^*$ and outputs a text $\boldsymbol{t} \in \mathcal{V}^*$.

- The watermark detection algorithm $\phi \leftarrow \mathsf{Detect}(\mathsf{pk}, \boldsymbol{t}')$. The deterministic algorithm Detect takes as input a public key pk, a candidate watermarked text $\boldsymbol{t}'$, and outputs a boolean value $\phi$, with $\phi = \mathtt{true}$ meaning valid and $\phi = \mathtt{false}$ meaning invalid.

**Properties.** A publicly detectable watermarking of LLM should satisfy the following properties. First property is completeness. The completeness property ensures that a text of sufficient length that was watermarked faithfully must be detected (i.e., must be treated as a valid watermarked text), except negligible probability. The second property is robustness. The robustness property requires that even if a watermarked text is modified, the embedded watermark cannot be eliminated and can still be detected. The second property is soundness. The soundness property requires that an adversary, after seeing multiple watermarked texts should not be able to generate a valid (i.e., detectable) but "different" watermarked text. The last property distortion-freeness is often used to describe the text

quality of watermarked text. Distortion-freeness ensures that the watermarking algorithm does not noticeably decrease the quality of the model output. We will give the formal definition of these properties in Appendix C.

# 3 SOUNDNESS AND ROBUSTNESS IN THE PRESENCE OF AN EDITING-ADVERSARY, AND AN IMPOSSIBILITY RESULT

As we discussed in the introduction, the conditions for robustness and soundness properties in Definition C.2 and Definition C.3 conflict with each other. Therefore, it is infeasible to achieve the two properties *simultaneously* based on the definitions in Fairoze et al. (2023). In this section, we will define the robustness and soundness properties in the presence of a new type of adversaries called *editing-adversaries*. We then formally prove an impossibility result of achieving the robustness and soundness properties at the same time in the presence of editing-adversaries. Jumping ahead, in Section 4, we will show to how to bypass the impossibility result by introducing a revised version of the definitions for robustness and soundness (in the presence of editing-adversaries).

## 3.1 WHY USING EDIT DISTANCE (INSTEAD OF OVERLAPPING SUBSTRING)

Using overlapping substrings to measure differences between two texts is equivalent to measuring text similarity by the length of the longest common substring. Compared to the length of overlapping substrings, *edit distance* has significant advantages in measuring text similarity. Unlike the length of overlapping substrings, edit distance evaluates the *minimum number of operations (insertions, deletions, and substitutions)* needed to transform one text into another. This allows it to comprehensively consider words change between two texts, whether these matching parts are successive or separated. Consequently, edit distance can more generally capture local similarities within texts, such as matching subsequences scattered in different positions, providing a more accurate similarity assessment.

## 3.2 SOUNDNESS AND ROBUSTNESS IN THE PRESENCE OF AN EDITING-ADVERSARY

To analyze if the two properties can be achieved simultaneously more formally, we redefine them with a unified parameter. The edit distance is commonly used to measure the dissimilarity between texts, making it a natural choice for describing the differences between the text generated by the adversary and the text generated by $\mathsf{Watermark}(\cdot)$. Because we use edit distance to describe the adversary's output, we refer to this type of adversary as an editing-adversary.

We use the edit distance $\mathsf{Distance}(t', t)$ to depict the relation between the output from the adversary and the original outputs. In addition, for a text $t$ and a set $\mathcal{Q}$ of texts where $\mathcal{Q} = \{t_1, t_2, \ldots, t_q\}$ and $q \in \mathbb{N}$, we define the edit distance between $t'$ and $\mathcal{Q}$ as $\mathsf{Distance}(t', \mathcal{Q}) = \min\{\mathsf{Distance}(t', t_i)\}_{t_i \in \mathcal{Q}}$.

**Definition 3.1** (**d**-Robustness)**.** We say publicly detectable watermarking scheme $\mathsf{PDWS} = (\mathsf{Setup}, \mathsf{Watermark}, \mathsf{Detect})$ is **d**-*robust*, if for all PPT editing-adversaries $\mathcal{A}$, for every prompt $\boldsymbol{\rho} \in \mathcal{V}^*$, it holds that

$$\Pr \left[ \begin{array}{l} (\mathsf{pk}, \mathsf{sk}) \xleftarrow{\$} \mathsf{Setup}(1^\lambda); t \xleftarrow{\$} \mathsf{Watermark}(\mathsf{sk}, \boldsymbol{\rho}); \\ t' \xleftarrow{\$} \mathcal{A}(\mathsf{pk}, t) \\ : (\mathsf{Detect}(\mathsf{pk}, t') = \texttt{false}) \bigwedge (\mathsf{Distance}(t', t) \le \mathbf{d}) \end{array} \right] \le \mathsf{negl}(\lambda).$$

**Definition 3.2** (**d**-Soundness)**.** We say publicly detectable watermarking scheme $\mathsf{PDWS} = (\mathsf{Setup}, \mathsf{Watermark}, \mathsf{Detect})$ is **d**-*sound*, if for all PPT editing-adversaries $\mathcal{A}$, it holds that

$$\Pr \left[ \begin{array}{l} (\mathsf{pk}, \mathsf{sk}) \xleftarrow{\$} \mathsf{Setup}(1^\lambda); t' \xleftarrow{\$} \mathcal{A}^{\mathsf{Watermark}(\mathsf{sk}, \cdot)}(\mathsf{pk}) \\ : (\mathsf{Detect}(\mathsf{pk}, t') = \texttt{true}) \bigwedge (\mathsf{Distance}(t', \mathcal{Q}) \ge \mathbf{d}) \end{array} \right] \le \mathsf{negl}(\lambda),$$

where $\mathcal{Q}$ is the history of queries that the editing-adversary $\mathcal{A}$ made to the watermarking oracle $\mathsf{Watermark}(\mathsf{sk}, \cdot)$.

The parameter **d** quantifies the extent to which the adversary alters the watermarked text. This parameter constrains the difference between the original text $t$ and the manipulated text $t'$. By using a unified parameter **d** for both robustness and soundness, we can analyze whether the protocol can simultaneously satisfy these two properties.

## 3.3 AN IMPOSSIBILITY RESULT

In order to prove the impossibility result we first define **single** watermarking scheme.

**Definition 3.3.** For a publicly-detectable watermarking scheme PDWS in Definition 2.3, if the output $\phi \leftarrow \mathsf{Detect}(\mathsf{pk}, t')$ is a single boolean value, we say PDWS is a publicly-detectable single watermarking scheme.

**Theorem 3.4** (Impossibility of achieving **d**-robustness and **d**-soundness simultaneously). *Let* PDWS = (Setup, Watermark, Detect) *be a publicly detectable* **single** *watermarking scheme, then* PDWS *cannot achieve* **d**-*robustness and* **d**-*soundness simultaneously.*

We leave the proof of Theorem 3.4 in Appendix D. Theorem 3.4 shows that if the PDWS is a single watermarking scheme, then it cannot achieve **d**-*robustness* and **d**-*robustness* simultaneously. We also show the impossibility for substring-adversaries as in Fairoze et al. (2023) in Theorem D.1 in the Appendix.

# 4 PUBLICLY-DETECTABLE DUAL WATERMARKING: DEFINITIONS

## 4.1 SYNTAX

In order to achieve **d**-*robustness* and **d**-*soundness* simultaneously, we define publicly-detectable dual watermarking scheme. The primary distinction from the original publicly-detectable single watermarking scheme is that the $\mathsf{Detect}(\cdot)$ algorithm will output a tuple of boolean values, with one serving the robustness property and the other the soundness property. We will highlight the difference in blue in this section.

**Definition 4.1** (Publicly-Detectable Dual Watermarking Scheme). A publicly detectable watermarking scheme PD2WS for an auto-regressive model Model over token vocabulary $\mathcal{V}$ consists of a tuple of algorithms PD2WS = (Setup, Watermark, Detect) where:

— The setup algorithm $(\mathsf{pk}, \mathsf{sk}) \xleftarrow{\$} \mathsf{Setup}(1^\lambda)$.

   The algorithm Setup takes as input a security parameter $1^\lambda$ and outputs a pair of public and private keys $(\mathsf{pk}, \mathsf{sk})$.

— The watermarking algorithm $t \xleftarrow{\$} \mathsf{Watermark}(\mathsf{sk}, \rho)$.

   The algorithm Watermark takes as input a private key sk and a prompt $\rho \in \mathcal{V}^*$ and outputs a text $t \in \mathcal{V}^*$.

— The watermark detection algorithm $\langle \phi_r, \phi_s \rangle \leftarrow \mathsf{Detect}(\mathsf{pk}, t')$.

   The deterministic algorithm $\mathsf{Detect}(\cdot)$ takes as input a public key pk, a candidate watermarked text $t'$, and outputs a tuple of boolean values $\langle \phi_r, \phi_s \rangle$. If $\phi_r = \texttt{true}$ the robustness watermark is detected. If $\phi_s = \texttt{true}$ the soundness watermark is detected.

## 4.2 PROPERTIES

Distortion-freeness is independent of the $\mathsf{Detect}(\cdot)$ algorithm, requiring no additional modifications, which we will not delve into here. We revise the definitions of completeness, robustness, and soundness below, emphasizing the distinctions in blue.

**Definition 4.2** ($\gamma$-Completeness). We say publicly detectable dual watermarking scheme PD2WS = (Setup, Watermark, Detect) is $\gamma$-*complete*, if for every prompt $\rho \in \mathcal{V}^*$, it holds that

$$\Pr \left[ \begin{array}{l} (\mathsf{pk}, \mathsf{sk}) \xleftarrow{\$} \mathsf{Setup}(1^\lambda); t \xleftarrow{\$} \mathsf{Watermark}(\mathsf{sk}, \rho); \\ \langle \phi_r, \phi_s \rangle \leftarrow \mathsf{Detect}(\mathsf{pk}, t) \\ : ((\phi_r = \texttt{false}) \vee (\phi_s = \texttt{false})) \wedge (|t| \geq \gamma) \end{array} \right] \leq \mathsf{negl}(\lambda).$$

**Definition 4.3** (**d**-Robustness). We say publicly detectable dual watermarking scheme PD2WS = (Setup, Watermark, Detect) is **d**-*robust*, if for all PPT editing-adversaries $\mathcal{A}$, for every prompt $\rho \in$

$\mathcal{V}^*$, it holds that

$$\Pr\left[\begin{array}{l}(\mathsf{pk},\mathsf{sk}) \xleftarrow{\$} \mathsf{Setup}(1^\lambda); \boldsymbol{t} \xleftarrow{\$} \mathsf{Watermark}(\mathsf{sk},\boldsymbol{\rho}); \\ \boldsymbol{t}' \xleftarrow{\$} \mathcal{A}(\mathsf{pk},\boldsymbol{t}); \langle\phi_r,\phi_s\rangle \leftarrow \mathsf{Detect}(\mathsf{pk},\boldsymbol{t}') \\ : (\phi_r = \texttt{false}) \wedge (\mathsf{Distance}(\boldsymbol{t},\boldsymbol{t}') \leq \mathbf{d})\end{array}\right] \leq \mathsf{negl}(\lambda).$$

**Definition 4.4 (d-Soundness).** We say publicly detectable dual watermarking scheme PD2WS = (Setup, Watermark, Detect) is $\boldsymbol{d}$-*sound*, if for all PPT editing-adversaries $\mathcal{A}$, it holds that

$$\Pr\left[\begin{array}{l}(\mathsf{pk},\mathsf{sk}) \xleftarrow{\$} \mathsf{Setup}(1^\lambda); \boldsymbol{t}' \xleftarrow{\$} \mathcal{A}^{\mathsf{Watermark}(\mathsf{sk},\cdot)}(\mathsf{pk}); \\ \langle\phi_r,\phi_s\rangle \leftarrow \mathsf{Detect}(\mathsf{pk},\boldsymbol{t}') \\ : (\phi_s = \texttt{true}) \wedge (\mathsf{Distance}(\boldsymbol{t}',\mathcal{Q}) \geq \mathbf{d})\end{array}\right] \leq \mathsf{negl}(\lambda),$$

where $\mathcal{Q}$ is the history of queries that the editing-adversary $\mathcal{A}$ made to the watermarking oracle Watermark(sk, $\cdot$).

# 5 PUBLICLY-DETECTABLE DUAL WATERMARKING: CONSTRUCTION

In this section, we show how to bypass the impossibility result as we demonstrated in the previous section. Due to space limitations, we provide only a brief description of the construction in the main text, with the complete version included in Appendix E. Our novel construction which is named as Publicly-Detectable Dual Watermarking Scheme (PD2WS) will utilize **two** different watermarking strategies, *short-range watermarking* and *long-range watermarking*, for generating text of a LLM. Short-range watermarking means that when a word in text $\boldsymbol{t}$ is modified, it only impacts a small number of bits (at least 1 bit) in the extracted watermark. This ensures that even if certain words are modified, the extracted watermark remains similar to the original. Short-range watermarking provides the robustness property. On the other hand, long-range watermarking means that when a word is modified, it will affect a lot of bits in the extracted watermark. This implies that when a few words are modified, the extracted watermark is broken. Long-range watermarking provides the soundness property.

To embed watermark information in tokens, it is essential to select suitable tokens to signify 0 and 1 individually. We utilize the least significant bit of the hash value of a token to indicate the respective bit of the embedded watermark. The study in Kirchenbauer et al. (2023) has demonstrated that employing a modified softmax function can enhance the likelihood of selecting appropriate tokens with minimal effect on text quality. We use a similar method to generate a token. The algorithm TGPB takes prompt $\boldsymbol{\rho}$, previous output tokens $\boldsymbol{t}$, a preferred bit $b$ and tune factor $\tau$ as input. TGPB first employ an auto-regressive model Model($\cdot$) to produce a vector of logits $\mathcal{D}$ of each word in the vocabulary $\mathcal{V}$. The procedure that the tokens are generated with dual watermarks is illustrated in Figure 1.

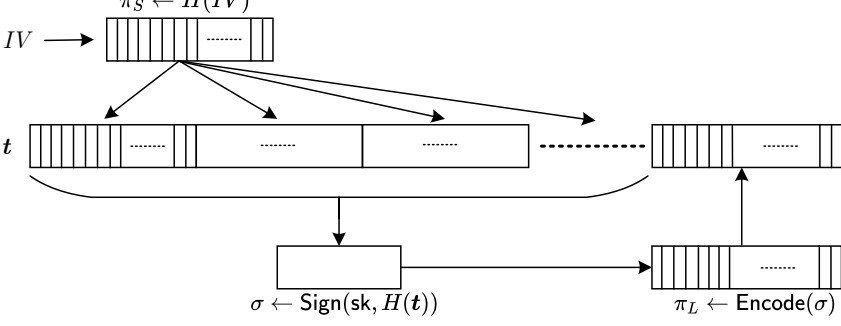

Figure 1: The short-range watermark $\pi_S$ is embedded in the tokens periodically for every $m$ tokens. The long-range watermark $\pi_L$ is embedded in the last $\ell$ tokens. All but the last $\ell$ tokens are used as input text of LWG to generate $\pi_L$.

The short-range watermark is embedded periodically in every $m$ token except the last $\ell$ tokens. As the generation of the short-range watermark is from a constant initial vector, the short-range watermark remains the same in each period. The generative model generates the sequence of tokens which are

embedded with the short-range watermark. The generation of the long-range watermark, on the other hand, depends on the tokens already generated which are embedded with the short-range watermark. The long-range watermark is only embedded once in the last $\ell$ tokens. In order to detect if a text $\boldsymbol{t}'$ contains the short-range watermark, all the substrings of $\boldsymbol{t}'$ will be checked. For one substring, each token is mapped to a bit using the hash function, thereby forming a bit string $\pi'_S$ of length $m$. Then the edit distance between $\pi_S$ and $\pi'_S$ is used to measure if $\pi'_S$ is a valid watermark where $\pi_S$ is the hash value of the public initial vector $IV$. If the edit distance is less than a predefined threshold $T$, then the output is true. The long-range watermark is embedded in the last $\ell$ tokens. ECC is used to recover the original signature $\sigma$ from $\pi_L$. The first $n - \ell$ tokens are used as the message to generate the signature $\sigma$. If the input text is not modified, the signature verification will return true.

We will examine the security characteristics of our publicly-detectable dual watermarking scheme, PD2WS. We can demonstrate that it achieves $\gamma$-Completeness, **d**-Robustness, and **d**-Soundness. Further details are available in Appendix F.

# 6 PUBLICLY-DETECTABLE DUAL WATERMARKING: IMPLEMENTATION AND EVALUATION

We implement our watermarking scheme using three publicly available LLMs : OPT-2.7B Zhang et al. (2022) , LLaMA-7B Touvron et al. (2023) and Mistral Jiang et al. (2023). Similar to previous works Kirchenbauer et al. (2023); Fairoze et al. (2023); Kuditipudi et al. (2023), we conducted our experiments using the news-like subset of the C4 dataset Raffel et al. (2020) as the prompt input.

## 6.1 PROBABILITY OF WATERMARK EMBEDDED

We first evaluate the probability that a watermark bit is embedded correctly in Algorithm 4. This probability is only related to the hash value of the token returned by LLM model, and this is not related to the model's performance. We will complete the following experiment using the OPT-2.7B Zhang et al. (2022) model as an example. As described in Algorithm 4, the distribution of each token is computed by a modified softmax function, the token with the highest probability is chosen to output. The probability that a correct watermark bit is embedded is tuned by the parameter $\tau$. If $\tau = 0$, the probability is decided by the original logits value of each token output from Model$(\cdot)$. The chosen token $x$ is independent of the preferred bit $b$. We have $\Pr[\text{LSB}(H(x)) = b] = \frac{1}{2}$. In this case, the preferred bit is embedded in the token correctly with probability $p_{\text{good}} = \frac{1}{2}$ which is low. In order to increase the probability that a token is embedded correctly. The modified softmax function tunes the probability with the parameter $\tau$. If a token $x$ satisfies that $\text{LSB}(H(x)) = b$ its probability will be increased, otherwise will be decreased correspondingly.

In order to determine how the parameter $\tau$ benefit a watermarking bit embedding correctly we observe the vector of logits $\mathcal{D}$ of tokens when Model$(\cdot)$ is called to generate a token. We use 5 different prompts and generate token vectors with the length of 100 for each prompt. The number of tokens of top 4 highest logits values are recorded as in Figure 2. The average of the highest logits value is about 20.05 which is 3.08 larger than the average of second highest logits value. If we set $\tau > 3.08$ then the second token will have good chance to be tried if the highest one $x$ does not satisfy $\text{LSB}(H(x)) = b$. The larger the parameter $\tau$ is, the more tokens have a chance to be tried.

We evaluated the probability that a preferred bit is not embedded correctly (bad probability) with $0 \leq \tau \leq 10$. For each $\tau$ we tried 2000 tokens with random preferred bit. We illustrate the bad probability according to the parameter $\tau$ in Figure 3.

## 6.2 TEXT QUALITY EVALUATION

The watermarking scheme will decrease the text quality and the watermark can be viewed as noise. Distortion-freeness in Definition C.4 ensures that the watermarking algorithm does not noticeably decrease the quality of the model output. In this paper, we do not analyze the distortion-freeness theoretically. We evaluate the text quality with experiments. Similar to the approach in Kirchenbauer et al. (2023), we utilize perplexity to measure the quality of the text after watermark embedding. Specifically, perplexity is computed by taking the logarithm of the probability of each token at every position and then averaging them. Perplexity (PPL) is defined as the exponential average negative

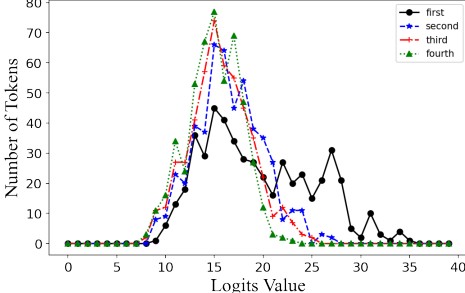

Figure 2: The top 4 logits values for token generation.

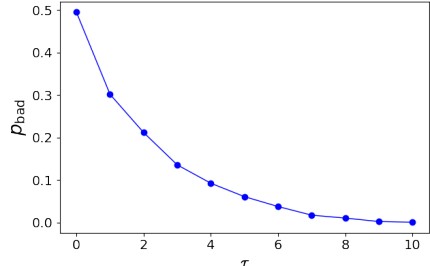

Figure 3: The bad probability over different $\tau$. When $\tau = 4$, the bad probability is about $10\%$. When $\tau = 8$, the bad probability is about $1\%$.

log-likelihood of a token sequence Chen et al. (1998). If we have a text $\boldsymbol{t} = (x_0, x_1, ..., x_t)$, the PPL of $\boldsymbol{t}$ is computed as $\mathrm{PPL}(x) = \exp\left\{-\frac{1}{t}\sum_{i=0}^{t}\log p(x_i|x_{<i})\right\}$. Here, $\log p(x_i|x_{<i})$ represents the log-likelihood of the $i$-th token conditioned on the preceding tokens $x_{<i}$.

This metric can be understood as the average number of options the model considers when predicting the next word. A lower perplexity value on a given test set indicates a better output quality. For large language models, beam search is commonly employed during text generation to enhance the quality of the generated output. The perplexity values for generated text typically range from 1.5 to 20 Zhao et al. (2023).

Our text quality evaluation utilized OPT-2.7B , LLaMA-7B and Mistral to compute perplexity. In order to evaluate how the parameter $\tau$ affects the text quality. We randomly chose 20 test prompts from C4 dataset for $0 \leq \tau \leq 10$ and conducted the experiment. The result is il-

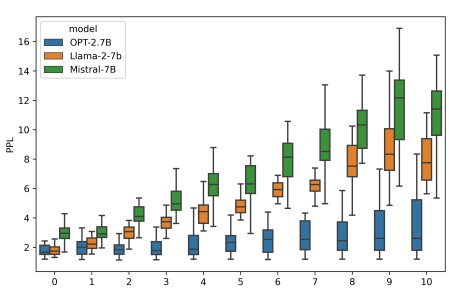

Figure 4: The PPL will increase when the tune factor $\tau$ increases.

lustrated in Figure 4. It can be observed that the perplexity of watermarked text increases as $\tau$ increases. This indicates that the text quality will decrease when the watermark is embedded with a higher probability. Our scheme can take a proper $\tau$ to embed the watermark correctly with a high probability while the text quality is good enough.

In Appendix G, we provide examples illustrating the example of text completions with different $\tau$.

### 6.3 A CONCRETE EXAMPLE OF PARAMETERS

Here, we provide a specific set of parameters for the scheme as an example to demonstrate that distance-soundness and distance-robustness can be simultaneously achieved. We employ a BLS signature scheme with a 48-byte signature length (384 bits). An error correction code ECC is utilized, where the input length is 384 bits and the output length is $\ell = 512$ bits, corresponding to the length of the long-range watermark $\ell = 512$. Assuming the SHA-256 hash function is employed to create a short-range watermark with a length of $m = 256$ bits. We establish the total length of the text generated from LLM as $n = 2048$ bits, indicating that there are $\frac{n-\ell}{m} = 6$ short-range watermarks embedded.

The error correction code ECC has a redundancy of 128 bits, allowing it to correct a maximum of $d = 64$ bits of errors. By setting the tuning factor as $\tau = 4$, we achieve $p_{\mathsf{bad}} = 0.1$. When $\mu = 0.1$, we ensure that $d \geq (1 + \mu) \cdot \ell \cdot p_{\mathsf{bad}} \approx 57$, guaranteeing the correction of errors in the long-range watermark with a high probability. If $\frac{n-\ell}{\ell}\left(\frac{T}{m} - (1 + \mu) \cdot p_{\mathsf{bad}}\right) > 1$, then we obtain $T > 114$. Setting the threshold as $T = 115$, we find that $\mathbf{d} = \frac{n-\ell}{m}T - (1 + \mu)(n - \ell) \cdot p_{\mathsf{bad}} \approx 515$, which simplifies

to $\mathbf{d} = 515$. We confirm that this specific set of parameters will achieve both $\mathbf{d}$-soundness and $\mathbf{d}$-robustness with a value of $\mathbf{d} = 515$.

Firstly, it is clear that $\mathbf{d} > \ell$. For any altered $\boldsymbol{t}'$ and query history $\mathcal{Q}$, if $\mathsf{Distance}(\boldsymbol{t}', \mathcal{Q}) \geq \mathbf{d}$, then $\boldsymbol{t}'$ must contain distinct tokens prior to the last $\ell = 512$ tokens compared to any $\boldsymbol{t} \in \mathcal{Q}$. The long-range watermarking detector will return `false` for the input $\boldsymbol{t}'$, ensuring the soundness property. Secondly, within the first 1536 tokens, 6 segments of tokens are embedded with a short-range watermark. In the case of any altered $\boldsymbol{t}'$ and an output text $\boldsymbol{t}$ generated by LLM, if $\mathsf{Distance}(\boldsymbol{t}', \boldsymbol{t}) \leq \mathbf{d}$, it implies that at least one segment of $\boldsymbol{t}'$ has an edit distance from the corresponding segment of $\boldsymbol{t}$ that is less than $\frac{\mathbf{d}}{6} \approx 86$. For this specific segment, the error bits of the embedded watermark are expected to be less than $m \cdot (1 + \mu) p_{\mathsf{bad}} \approx 29$ with a high probability. When considering these factors collectively, the distance of the extracted watermark from this segment compared to the short-range watermark is less than $86 + 29 = T$ with a high probability. Consequently, the long-range watermarking detector will return `true` for $\boldsymbol{t}'$ as input, ensuring the robustness property.

## 7 RELATED WORK

***AI-generated content detection.*** Early approaches to detecting AI-generated text typically involve identifying special features present in human-generated textLavergne et al. (2008); Beresneva (2016); Gehrmann et al. (2019). Deep learning is utilized as a binary classifier for this purpose in Zellers et al. (2019); Mitchell et al. (2023); Hendrik Kirchner et al. (2023). Another method involves fine-tuning pre-trained language models, as discussed in Wu et al. (2023); Liu et al. (2022). Research in Chakraborty et al. (2023) indicates that as AI-generated text approaches human quality, text distinguishers require longer text samples. Furthermore, research has demonstrated the possibility of training models to alter text in a way that deceives text distinguishers Krishna et al. (2023); Sadasivan et al. (2023).

***Watermarking for LLM-generated content.*** Recent research has explored the use of machine learning for watermarking, as evidenced by works such as Abdelnabi & Fritz (2021); Qiang et al. (2023); Yoo et al. (2023); Munyer & Zhong (2023); Liu et al. (2023). These schemes are purely empirical and lack of formal definition of security properties such as robustness, soundness, or distortion-freeness. In Kirchenbauer et al. (2023), it is demonstrated that a watermark can be inserted into the output of LLM if the model entropy is high.This study quantifies the distortions introduced by the watermark through the measurement of perplexity. In Kuditipudi et al. (2023), a family of watermarking schemes are developed to maximize robustness. The formal security properties such as soundness, completeness of LLM are defined in Christ et al. (2023). In Fairoze et al. (2023), the concept of publicly detectable schemes is explored for the first time. The robustness and soundness of this scheme are demonstrated under the assumption of substring overlapping.

***Some recent related work.*** The term "dual watermarking" has also been employed in Zhu et al. (2024) (a work parallel to ours). It optimize the efficiency and quality of watermarking by incorporating dual secret patterns into both the token probability distribution and sampling strategies. In a very recent paper Zhou et al. (2024), the authors noted that existing LLM watermarking schemes cannot simultaneously achieve robustness and soundness. This work aligns with the impossibility theorem presented in our paper. In recent study Zhang et al. (2023), the impossibility of achieving strong watermarks for generative models is proved. We remark, there is *no conflict* between the impossibility result and our feasibility result. We *assume* that the edit distance of text is bounded, and the attacker *is not allowed to change* the text a lot.

Due to space limitations, the details of related work is included in Appendix A.

## 8 CONCLUSION

In this paper, our focus is on watermarking techniques for LLMs. We define the security properties of a watermarking scheme based on edit distance and demonstrate the impossibility of achieving robustness and soundness simultaneously for a publicly-detectable single watermarking scheme.

Our major result is a new concept of *publicly-detectable dual watermarking scheme*. We propose a concrete construction, and then prove the security properties of the proposed scheme; Finally, we evaluate the critical parameters through experiments.

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

Appendix

## A  RELATED WORK

***AI-generated content detection.*** AI-generated content detection means that the content does not introduce any extra information when it is generated. The content is detected passively.

Early approaches to detecting AI-generated text typically involve identifying special features present in human-generated text. If these features are identified, it is considered to be generated by a human; otherwise, it is attributed to AI. Examples of such features include relative entropy scoring Lavergne et al. (2008), perplexity Beresneva (2016), and other statistical signals Gehrmann et al. (2019).

To automatically detect AI-generated text, researchers have proposed training-based classifiers. Deep learning is utilized as a binary classifier for this purpose in Zellers et al. (2019); Mitchell et al. (2023); Hendrik Kirchner et al. (2023). Another method involves fine-tuning pre-trained language models, as discussed in Wu et al. (2023); Liu et al. (2022). The issue with this approach is its reliance on the assumption that AI-generated text cannot mimic human-generated text with similar features. While this may hold for early AI models, as models improve, the distinct features of AI-generated text will diminish. For instance, GPT-4 OpenAI (2023) and other state-of-the-art models closely resemble human writing. Research in Chakraborty et al. (2023) indicates that as AI-generated text approaches human quality, text distinguishers require longer text samples.

Furthermore, research has demonstrated the possibility of training models to alter text in a way that deceives text distinguishers Krishna et al. (2023); Sadasivan et al. (2023).

***Watermarking for LLM-generated content.*** Watermarking hides identifying information within AI-generated text, enabling the detection of whether the text is AI-generated. Recent research has explored the use of machine learning for watermarking, as evidenced by works such as Abdelnabi & Fritz (2021); Qiang et al. (2023); Yoo et al. (2023); Munyer & Zhong (2023); Liu et al. (2023). However, it is important to note that all schemes in this category are purely empirical and lack of formal definition of security properties such as robustness, soundness, or distortion-freeness. Recently, a series of research have advanced the rigorous definition and security proof of LLM watermarking, and our work is also following this line of development. The main references are listed in the following.

In Kirchenbauer et al. (2023), it is demonstrated that a watermark can be inserted into the output of LLM if the model entropy is high. A watermark can be planted by hashing previous tokens to embed a watermark signal in the next token. Furthermore, this study quantifies the distortions introduced by the watermark through the measurement of perplexity, which reflects the difference between the distribution produced by the unaltered model and the distribution produced by the model with watermarking.

Another approach to LLM watermarking is the Gumbel softmax scheme introduced in Aaronson (2023). This scheme utilizes exponential minimum sampling to draw samples from the model using randomness derived from previous tokens (via hashing). Additionally, Kuditipudi et al. (2023) has developed a family of watermarking schemes that are designed to maximize robustness.

The formal security properties such as soundness, completeness of LLM are defined in Christ et al. (2023). The security properties are proved under the assumption that an contiguous substring of the output remaining sufficiently high entropy. The watermark in Christ et al. (2023) is undetectable without a secret key.

In Fairoze et al. (2023), the concept of publicly detectable schemes is explored for the first time. The scheme proposed in Fairoze et al. (2023) utilizes digital signatures to facilitate the public detection of the watermark. The robustness and soundness of this scheme are demonstrated under the assumption of substring overlapping. However, it is observed that the assumptions underlying these two properties are contradictory and *cannot be simultaneously satisfied*, as we discussed in Section 3. To circumvent the impossibility result, we introduce a novel watermarking approach termed "dual watermarking," detailed in Section 4. The concept of "dual watermarking" involves the use of two distinct watermarks to ensure robustness and soundness, respectively.

The term "dual watermarking" has also been employed in Zhu et al. (2024) (a work parallel to ours). The objective of the method presented in Zhu et al. (2024) is to optimize the efficiency and quality of watermarking by incorporating dual secret patterns into both the token probability distribution and sampling strategies. It is important to note that the design and security aspects explored in Zhu et al. (2024) are entirely distinct from those in our study.

The watermarking mechanism for generative models is still in the early stages of research. In recent study Zhang et al. (2023), the impossibility of achieving strong watermarks for generative models is proved. A strong watermarking scheme satisfies the property that a computationally bounded attacker cannot erase the watermark without causing significant quality degradation. In their paper, the authors demonstrated the attack on several existing watermarking schemes with minor quality degradation. However, their attack requires extra computing resources to alter tokens of text. We remark, there is *no conflict* between the impossibility result in Zhang et al. (2023), and our feasibility result (i.e., our dual watermarking in Section 5 and Section F): in our feasibility result, we *assume* that the edit distance of text is bounded, and the attacker *is not allowed to change* the text a lot.

In a very recent paper Zhou et al. (2024), the authors noted that existing LLM watermarking schemes cannot simultaneously achieve robustness and soundness, meaning they cannot resist both removal attacks and spoofing attacks at the same time. In their paper, they proposed a scheme called Bilevel, which uses two watermarking mechanisms to resist these two types of attacks separately. This work aligns with the impossibility theorem presented in our paper, and the constructed scheme also meets the definition of a Publicly-Detectable Dual Watermarking Scheme as provided in our paper. However, the paper does not provide a rigorous definition of security, nor does it specify the conditions required to achieve both security features simultaneously. The construction of the Bilevel scheme also has shortcomings. For instance, its digital signature-based approach requires the signature to be embedded strictly correctly into the output text, which in some cases may necessitate choosing tokens that significantly degrade text quality.

## B DETAILED PRELIMINARIES

### B.1 HASH FUNCTIONS

Our construction uses cryptographic hash functions $H : \{0,1\}^* \to \{0,1\}^m$, with $m$-bit output where $m \in \mathbb{N}$. In our security analysis, cryptographic hash functions will be treated as random oracles. As formalized in Bellare & Rogaway (1993) by Bellare and Rogaway, a random oracle is a random function drawn from the set of all possible functions uniformly and randomly (over specific input and output domains). We use $\mathrm{LSB}(H(x))$ to denote the *least significant bit* of $H(x)$.

### B.2 DIGITAL SIGNATURE SCHEMES

In our construction, we use a digital signature scheme to generate watermark sequences that will be embedded in the output tokens. Below, we present the definition of digital signature schemes; Please also see Katz & Lindell (2007).

**Definition B.1** (Digital Signature Scheme). A digital signature scheme consists of three PPT algorithms (Gen, Sign, Verify) such that:

- The key-generation algorithm $(\mathsf{pk}, \mathsf{sk}) \xleftarrow{\$} \mathsf{Gen}(1^\lambda)$.

  The algorithm Gen takes as input a security parameter $1^\lambda$ and outputs a pair of public and private keys $(\mathsf{pk}, \mathsf{sk})$.

- The signing algorithm $\sigma \xleftarrow{\$} \mathsf{Sign}(\mathsf{sk}, m)$.

  The algorithm Sign takes as input a private key $\mathsf{sk}$ and a message $m$ from some message space (that may depend on $\mathsf{pk}$), and outputs a signature $\sigma$.

- The verification algorithm $\phi \leftarrow \mathsf{Verify}(\mathsf{pk}, m, \sigma)$.

The deterministic algorithm Verify takes as input a public key pk, a message $m$, and a signature $\sigma$, and outputs a boolean value $\phi$, with $\phi = \texttt{true}$ meaning valid and $\phi = \texttt{false}$ meaning invalid.

**Definition B.2** (Completeness). We say digital signature scheme $\Sigma = (\mathsf{Gen}, \mathsf{Sign}, \mathsf{Verify})$ is *complete* if for any message $m$, it holds that

$$\Pr\left[\begin{array}{l} (\mathsf{pk}, \mathsf{sk}) \stackrel{\$}{\leftarrow} \mathsf{Gen}(1^\lambda); \sigma \stackrel{\$}{\leftarrow} \mathsf{Sign}(\mathsf{sk}, m) \\ : (\mathsf{Verify}(\mathsf{pk}, m, \sigma) = \texttt{true}) \end{array}\right] \geq 1 - \mathsf{negl}(\lambda).$$

**Definition B.3** (Unforgeability). We say digital signature scheme $\Sigma = (\mathsf{Gen}, \mathsf{Sign}, \mathsf{Verify})$ is *unforgeable* if for all PPT adversary $\mathcal{A}$, it holds that

$$\Pr\left[\begin{array}{l} (\mathsf{pk}, \mathsf{sk}) \stackrel{\$}{\leftarrow} \mathsf{Gen}(1^\lambda); (m^*, \sigma^*) \stackrel{\$}{\leftarrow} \mathcal{A}^{\mathsf{Sign}(\mathsf{sk}, \cdot)}(\mathsf{pk}) \\ : (\mathsf{Verify}(\mathsf{pk}, m^*, \sigma^*) = \texttt{true}) \bigwedge ((m^*, \sigma^*) \notin \mathcal{Q}) \end{array}\right] \leq \mathsf{negl}(\lambda),$$

where $\mathcal{Q}$ is the history of queries that the adversary $\mathcal{A}$ made to signing oracle $\mathsf{Sign}(\mathsf{sk}, \cdot)$.

### B.3 ERROR CORRECTING CODE

An error-correcting code (ECC) is a coding scheme used for the transmission of messages. In our construction, we utilize Error Correcting Code (ECC) to correct errors in watermark data. We remark that, in the context of AI-generated content, in Fairoze et al. (2023), the authors has already mentioned that ECC can be used for watermarking the LLM-generated text. The ECC encoding and decoding algorithms are defined as follows.

**Definition B.4** (Error Correcting Code). An error-correcting code ECC consists of a tuple of algorithms $\mathsf{ECC} = (\mathsf{Encode}, \mathsf{Decode})$.

- $\mathsf{c} \leftarrow \mathsf{Encode}(\mathsf{m})$. The Encode algorithm takes a message $\mathsf{m} \in \mathcal{M}$ as input and outputs $\mathsf{c}$ as a codeword.

- $\mathsf{m} \leftarrow \mathsf{Decode}(\mathsf{c}')$. The Decode algorithm recovers the original message from the received codeword $\mathsf{c}'$ which may have maximum distance $t$ from an original codeword $\mathsf{c}$.

The notation $[n, k, d]$ is used to present the parameters of ECC, where $n$ is the length of $\mathsf{c}$, $k$ is the length of $\mathsf{m}$ and $d$ is the minimal distance between any two different codewords. An error-correcting code can correct $t < \frac{d-1}{2}$ bits of errors at most.

### B.4 EDIT DISTANCE

Measuring the similarity between two strings is a crucial task in various domains. The *edit distance* (also known as the Levenshtein distance Levenshtein (1966)) is a commonly employed similarity measurement, which quantifies the minimum number of operations required to transform one string into another (i.e., insertion, deletion, and substitution). We use edit distance to limit how a text $\boldsymbol{t}$ can be modified by adversary.

Consider a finite alphabet set $\mathcal{V}$ whose elements are used to construct strings. Let $Z_I$, $Z_D$ and $Z_S$ be finite sets of integers. Let the function $I : \mathcal{V} \to Z_I$ be the *insertion cost* function, i.e., $I(a)$ is the cost of inserting the element $a \in \mathcal{V}$ into a given string. Similarly, define the *deletion cost* function as $D : \mathcal{V} \to Z_D$ so that $D(a)$ is the cost of deleting the element $a \in \mathcal{V}$ from a given string. Finally, define the *substitution cost* function $S : \mathcal{V} \times \mathcal{V} \to Z_S$ so that for $a, b \in \mathcal{V}$, $S(a, b)$ is the cost of replacing the element a by the element b in a given string.

Given two strings of length $m$ and $n$, denoted by $\boldsymbol{t} \in \mathcal{V}^m$ and $\boldsymbol{t}' \in \mathcal{V}^n$ respectively, consider the sequence of insertion, deletion and substitution operations needed to transform $\boldsymbol{t}$ into $\boldsymbol{t}'$ and the corresponding aggregate cost of the transformation. The edit distance between $\boldsymbol{t}$ and $\boldsymbol{t}'$ is defined as the minimum aggregate cost of transforming $\boldsymbol{t}'$ into $\boldsymbol{t}$ which is denoted as $\mathsf{Distance}(\boldsymbol{t}', \boldsymbol{t})$. The general definition of edit distance given above considers different weights for different operations.

In this paper, we will consider a simpler definition which is given below.

**Definition B.5.** For all $a, b \in \mathcal{V}$, let $I(a) = D(a) = 1$, $S(a, b) = 1$ when $a \neq b$, and $S(a, a) = 0$. Then, the edit distance is defined as the minimum number of insertion, deletion and substitution operations required to convert $\boldsymbol{t}'$ into $\boldsymbol{t}$.

**Calculation for Edit Distance**  Consider two texts $t$ and $t'$. First, we parse $t$ into $(x_1, x_2, \ldots, x_m)$ where $x_i \in \mathcal{V}$ for all $i \in \{1, \ldots, m\}$. Similarly, we parse $t'$ into $(x'_1, x'_2, \ldots, x'_n)$ where $x'_j \in \mathcal{V}$ for all $j \in \{1, \ldots, n\}$. We use $M(i, j)$ to denote the edit distance between the two substrings $\hat{t} = x_1, x_2, \ldots, x_i$ and $\hat{t}' = x'_1, x'_2, \ldots, x'_j$. The problem of finding the edit distance between $t$ and $t'$ can be solved in $O(mn)$ time via dynamic programming Gusfield. (1997).

Let $M(0, 0) = 0$, for $1 \leq i \leq m$, $1 \leq j \leq n$, define $M(i, 0) = \sum_{k=1}^{i} I(x_k)$, and $M(0, j) = \sum_{k=1}^{j} D(x'_k)$. Then, the edit distance $M(m, n)$ is defined by the following recurrence relation for $1 \leq i \leq m$, $1 \leq j \leq n$:

$$M(i, j) = \min \left\{ \begin{array}{c} M(i-1, j) + D(x'_j), \\ M(i, j-1) + I(x_i), \\ M(i-1, j-1) + S(x_i, x'_j) \end{array} \right\}.$$

For convenience, we use $\mathbf{d} = \mathsf{Distance}(t, t') = M(m, n)$ to denote the edit distance between $t$ and $t'$ with the length of $m,n$ respectively.

### B.5  CHERNOFF BOUND

There are many different forms of Chernoff bounds with different assumptions. We use a simple case of a sum of independent Bernoulli trials. In a Bernoulli trial the random variable only takes the value 1 with probability $p$ and value 0 with probability $1 - p$.

**Theorem B.6.** *Let $X = \sum_{i=1}^{n} X_i$, where $X_i = 1$ with probability $p > 0$ and $X_i = 0$ with probability $1 - p$, and all $X_i$ are independent. Let $\mu = \mathbb{E}(X) = n \cdot p$. For all $0 < \delta < 1$, we have*

(i) Upper Tail: $\Pr[(X \geq (1 + \delta)\mu)] \leq e^{-\delta^2 \mu / 3} = e^{-\Omega(n)}$;

(ii) Lower Tail: $\Pr[(X \leq (1 - \delta)\mu)] \leq e^{-\delta^2 \mu / 2} = e^{-\Omega(n)}$.

### B.6  SOFTMAX FUNCTION

The softmax function takes a vector $\mathbf{z}$ of $k$ real numbers as input and normalizes it into a probability distribution of $k$ probabilities that are proportional to the exponential of the input numbers. The original components of $\mathbf{z}$ can have any values and may not sum to 1. Upon applying softmax, each component will be in the range $(0, 1)$, with the sum of components equaling 1, enabling interpretation as probabilities. Moreover, higher input components will correspond to higher probabilities.

Softmax is significant for assigning a probability value to each element in a vector, indicating the likelihood of that element, instead of merely identifying one element as the maximum value in the vector. The Softmax function is commonly used in deep learning classification tasks. The softmax function $\mathbf{Softmax}(z_i)$ for $z_i \in \mathbf{z}$ is defined by the formula:

$$\mathbf{Softmax}(z_i) = \frac{\exp(z_i)}{\sum_{j=1}^{k} \exp(z_j)}.$$

## C  MATERIALS SUPPORTING DEFINITION

### C.1  PROPERTIES

First, we define the completeness; basically, the completeness property ensures that a text of sufficient length that was watermarked faithfully must be detected (i.e., must be treated as a valid watermarked text), except negligible probability.

**Definition C.1** ($\gamma$-Completeness)**.** We say publicly detectable watermarking scheme $\mathsf{PDWS} = (\mathsf{Setup}, \mathsf{Watermark}, \mathsf{Detect})$ is $\gamma$-*complete* if for every prompt $\rho \in \mathcal{V}^*$, it holds that

$$\Pr \left[ \begin{array}{l} (\mathsf{pk}, \mathsf{sk}) \xleftarrow{\$} \mathsf{Setup}(1^\lambda); t \xleftarrow{\$} \mathsf{Watermark}(\mathsf{sk}, \rho) \\ : (\mathsf{Detect}(\mathsf{pk}, t) = \mathtt{false}) \bigwedge (|t| \geq \gamma) \end{array} \right] \leq \mathsf{negl}(\lambda).$$

We now describe the robustness and soundness properties as in Fairoze et al. (2023). Intuitively, the robustness property requires that even if a watermarked text is modified, the embedded watermark cannot be eliminated and can still be detected. However, an adversary could simply remove the entire watermarked text so that the embedded watermark can be eliminated. To avoid this trivial attack, in the formalization for the robustness property in Fairoze et al. (2023), the adversary is not allowed to remove the entire watermarked text; instead, the modified version from the adversary, denoted as $t'$, and the original version of the watermarked text, denoted as $t$, must *share* at least a $\delta$-length segment, where $\delta \in \mathbb{N}$.

On the other hand, the soundness property requires that an adversary, after seeing multiple watermarked texts, say $t_1, t_2, \ldots, t_q$, should not be able to generate a valid (i.e., detectable) but "different" watermarked text. We will introduce some notations, and formally define the *difference* between two watermarked texts.

**Notations $\bowtie_n$ and $\not\bowtie_n$.** To facilitate our presentation, we introduce the notation "$\bowtie_n$" and its negation "$\not\bowtie_n$". Concretely, consider two texts $t, t' \in \mathcal{V}^*$. If the two texts $t'$ and $t$ share at least an $n$-length segment, we write $t' \bowtie_n t$. In contrast, if there is no overlapping of an $n$-length window between the two texts $t'$ and $t$, we write $t' \not\bowtie_n t$.

In addition, when the text $t'$ does not overlap an $n$-length window of tokens with any of the texts in a set $\mathcal{Q}$, where $\mathcal{Q} = \{t_1, t_2, \ldots, t_q\}$ and $q \in \mathbb{N}$, we write $(t' \not\bowtie_n t_1) \wedge (t' \not\bowtie_n t_2) \wedge \cdots \wedge (t' \not\bowtie_n t_q)$; when the context is clear, we also write $t' \not\bowtie_n \mathcal{Q}$.

We are now ready to formally define the robustness and soundness properties as in Fairoze et al. (2023). We remark that the adversaries are restricted in the sense that their behavior on a text can be defined with substring; we thus call them substring-adversaries.

**Definition C.2** ($\delta$-Robustness). We say publicly detectable watermarking scheme PDWS = (Setup, Watermark, Detect) is $\delta$-*robust* if for all PPT substring-adversaries $\mathcal{A}$, for every prompt $\rho \in \mathcal{V}^*$, it holds that

$$\Pr\left[\begin{array}{l} (\mathsf{pk}, \mathsf{sk}) \xleftarrow{\$} \mathsf{Setup}(1^\lambda); t \xleftarrow{\$} \mathsf{Watermark}(\mathsf{sk}, \rho); \\ t' \xleftarrow{\$} \mathcal{A}(\mathsf{pk}, t) : (\mathsf{Detect}(\mathsf{pk}, t') = \mathtt{false}) \bigwedge (t' \bowtie_\delta t) \end{array}\right] \le \mathsf{negl}(\lambda).$$

**Definition C.3** ($k$-Soundness). We say publicly detectable watermarking scheme PDWS = (Setup, Watermark, Detect) is $k$-*sound* if for all PPT substring-adversaries $\mathcal{A}$, it holds that

$$\Pr\left[\begin{array}{l} (\mathsf{pk}, \mathsf{sk}) \xleftarrow{\$} \mathsf{Setup}(1^\lambda); t' \xleftarrow{\$} \mathcal{A}^{\mathsf{Watermark}(\mathsf{sk}, \cdot)}(\mathsf{pk}) \\ : (\mathsf{Detect}(\mathsf{pk}, t') = \mathtt{true}) \bigwedge (t' \not\bowtie_k \mathcal{Q}) \end{array}\right] \le \mathsf{negl}(\lambda),$$

where $\mathcal{Q}$ is the history of queries that the substring-adversary $\mathcal{A}$ made to the watermarking oracle Watermark(sk, ·).

Distortion-freeness ensures that the watermarking scheme does not significantly degrade the quality of the text.

**Definition C.4** ($\epsilon$-Distortion-freeness). We say publicly detectable watermarking scheme PDWS = (Setup, Watermark, Detect) is $\epsilon$-*distortion-free* if for all PPT distinguishers $\mathcal{D}$, it holds that

$$\left| \Pr\left[\mathcal{D}^{\mathsf{Model}, \mathsf{GenModel}}(1^\lambda) = 1\right] - \Pr\left[\begin{array}{l} (\mathsf{pk}, \mathsf{sk}) \xleftarrow{\$} \mathsf{Setup}(1^\lambda) \\ : \mathcal{D}^{\mathsf{Model}, \mathsf{Watermark}(\mathsf{sk}, \cdot)}(1^\lambda) = 1 \end{array}\right] \right| \le \epsilon,$$

where $\epsilon \ge 0$.

# D    MATERIALS SUPPORTING IMPOSSIBILITY RESULT

## D.1    IMPOSSIBILITY WITH EDITING ADVERSARY

**Theorem 3.4** (Impossibility of achieving **d**-robustness and **d**-soundness simultaneously)**.** *Let* PDWS = (Setup, Watermark, Detect) *be a publicly detectable* **single** *watermarking scheme, then* PDWS *cannot achieve **d**-robustness and **d**-soundness simultaneously.*

*Proof.* Let $(\mathsf{pk}, \mathsf{sk})$ be a key pair which is generated as $(\mathsf{pk}, \mathsf{sk}) \leftarrow \mathsf{Setup}(1^\lambda)$. Let $\mathcal{Q}$ be the history of queries as $\mathcal{Q} \leftarrow \mathsf{Watermark}(\mathsf{sk}, \cdot)$. Let $\mathcal{A}$ be any PPT algorithm.

Assume that PDWS is **d**-sound. After obtaining $\mathcal{Q}$, the algorithm $\mathcal{A}$ produces an output $t' \leftarrow \mathcal{A}(\text{pk})$. The distance between $t'$ and $\mathcal{Q}$ is $\text{Distance}(t', \mathcal{Q}) = \mathbf{d}$, indicating that there exists a text $t \in \mathcal{Q}$ such that $\text{Distance}(t', t) = \mathbf{d}$. Following the Definition 3.2, we have

$$\Pr[\text{Detect}(\text{pk}, t') = \texttt{true}] \leq \text{negl}(\lambda). \tag{1}$$

Assume that PDWS is also **d**-robust. The text $t'$ which is generated by algorithm $\mathcal{A}$ as $t' \leftarrow \mathcal{A}(\text{pk}, t)$ satisfies $\text{Distance}(t', t) = \mathbf{d}$. Following the Definition 3.1, $\Pr[\text{Detect}(\text{pk}, t') = \texttt{false}] \leq \text{negl}(\lambda)$. That is

$$\Pr[\text{Detect}(\text{pk}, t') = \texttt{true}] \geq 1 - \text{negl}(\lambda). \tag{2}$$

Given that PDWS is publicly-detectable **single** watermarking scheme, the output of $\text{Detect}(\text{pk}, t') = \texttt{true}$ is a single boolean value so that $\Pr[\text{Detect}(\text{pk}, t') = \texttt{true}]$ must be the same value for robustness and soundness. Putting the equations (1) and (2) together we obtain

$$\text{negl}(\lambda) \geq \Pr[\text{Detect}(\text{pk}, t') = \texttt{true}] \geq 1 - \text{negl}(\lambda). \tag{3}$$

That is $\text{negl}(\lambda) \geq 1/2$ which is contradicted with the definition of negligible function. □

### D.2 IMPOSSIBILITY WITH SUBSTRING ADVERSARY

We will show that it is impossible to achieve $\delta$-robustness and $k$-soundness simultaneously which is defined in Fairoze et al. (2023).

**Theorem D.1** (Impossibility of achieving $\delta$-robustness and $k$-soundness simultaneously). *Let* PDWS = (Setup, Watermark, Detect) *be a publicly detectable* **single** *watermarking scheme, then* PDWS *cannot achieve $\delta$-robustness and $k$-soundness simultaneously with substring-adversaries $\mathcal{A}$.*

*Proof.* Let $\mathcal{Q}$ be the set of queries which are made by $\mathcal{A}$. Let $t \in \mathcal{Q}$ be a text which is generated by LLM. Let text $t'$ be the output which is generated by $\mathcal{A}$. Following the $\delta$-robustness in Definition C.2, the modified text $t'$ and the original watermarked text $t$ satisfies that $t' \bowtie_\delta t$, where $\delta \in \mathbb{N}$.

On the other hand, based on the $k$-robustness Definition C.3, the modified text $t'$ and the query history $\mathcal{Q}$ satisfies $t' \not\bowtie_k \mathcal{Q}$, where $k \in \mathbb{N}$.

Suppose that PDWS achieves $\delta$-robustness and $k$-soundness simultaneously. If $\delta \geq k$, there is no modified text $t'$ that can satisfy both $t' \bowtie_\delta t$ and $t' \not\bowtie_k t$. If $\delta < k$, suppose a modified text $t'$ satisfies that $t' \bowtie_\delta t$ and $t' \not\bowtie_k t$.

Given the robustness property holds, we have $\Pr[\text{Detect}(\text{pk}, t') = \texttt{false}] \leq \text{negl}(\lambda)$ which means $\Pr[\text{Detect}(\text{pk}, t') = \texttt{true}] \geq 1 - \text{negl}(\lambda)$. Given that the soundness property holds, we have $\Pr[\text{Detect}(\text{pk}, t') = \texttt{true}] \leq \text{negl}(\lambda)$ which contradicts with the fact that the robustness property also holds. This completes the proof. □

## E DETAILS OF PUBLICLY-DETECTABLE DUAL WATERMARKING CONSTRUCTION

In this section, we show how to bypass the impossibility result as we demonstrated in the previous section. Our novel construction which is named as Publicly-Detectable Dual Watermarking Scheme (PD2WS) will utilize **two** different watermarking strategies, *short-range watermarking* and *long-range watermarking*, for generating text of a LLM.

Short-range watermarking means that when a word in text $t$ is modified, it only impacts a small number of bits (at least 1 bit) in the extracted watermark. This ensures that even if certain words are modified, the extracted watermark remains similar to the original. Short-range watermarking provides the robustness property.

On the other hand, long-range watermarking means that when a word is modified, it will affect a lot of bits in the extracted watermark. This implies that when a few words are modified, the extracted watermark is broken. Long-range watermarking provides the soundness property.

Following the definition of publicly-detectable watermarking scheme in Definition 2.3, PD2WS contains three algorithms: $\mathsf{Setup}(\cdot)$, $\mathsf{Watermark}(\cdot)$ and $\mathsf{Detect}(\cdot)$. $\mathsf{Setup}(1^\lambda)$ utilizes the key-generation algorithm $\mathsf{Gen}(1^\lambda)$ of signature scheme to generate a pair of keys $(\mathsf{pk}, \mathsf{sk})$ which is simple. We introduce Watermark and Detect algorithms in the following two subsections respectively.

### E.1 DUAL WATERMARKING OF GENERATIVE MODELS

The $\mathsf{Watermark}(\mathsf{sk}, \boldsymbol{\rho})$ algorithm is implemented with three subroutines: watermark generation, watermark embedding and generative model of watermarking.

#### E.1.1 WATERMARK GENERATION

In our construction, the watermark is generated with public information and the private key which are input into the LLM as parameters. Watermarks for the two halves are generated separately.

**Short-range Watermark Generation** We define the short-range watermark generation algorithm abbreviated as SWG in Algorithm 2. The short-range watermark is the hash value of a public initial vector $IV$. The output of SWG is denoted as $\pi_S$ with the length of $m$ bits.

---

**Algorithm 2** Short-range Watermark Generation (SWG)

**Input:** $IV$

    $\pi_S \leftarrow H(IV)$
    return $\pi_S$

---

**Long-range Watermark Generation** We define the long-range watermark generation algorithm abbreviated as LWG in Algorithm 3. First, the signature $\sigma$ is generated by signing the hash value of the previous tokens. Then the signature is encoded with the error correcting code. The output of $\pi_L = \mathsf{Encode}(\sigma)$ is used as long-range watermark. The error correcting code will ensure that if the watermark is modified slightly the signature still can be recovered. The result of LWG is denoted as $\pi_L$ with the length of $\ell$ bits.

---

**Algorithm 3** Long-range Watermark Generation (LWG)

**Input:** $\boldsymbol{t}$, sk

    $\sigma \xleftarrow{\$} \mathsf{Sign}(\mathsf{sk}, H(\boldsymbol{t}))$
    $\pi_L \leftarrow \mathsf{Encode}(\sigma)$
    return $\pi_L$

---

#### E.1.2 PROBABILISTIC WATERMARK EMBEDDING

To embed watermark information in tokens, it is essential to select suitable tokens to signify $0$ and $1$ individually. We utilize the least significant bit of the hash value of a token to indicate the respective bit of the embedded watermark. In the absence of additional constraints, this token bit generated by a language model will match the watermark bit with a probability of $1/2$.

If the token selected by the LLM with the highest probability does not meet this criterion, alternative tokens must be explored. This approach contradicts the principle of selecting the token with the highest probability, and opting for alternative tokens could potentially degrade the quality of the output text. The study in Kirchenbauer et al. (2023) has demonstrated that employing a modified softmax function can enhance the likelihood of selecting appropriate tokens with minimal effect on text quality. The definition of softmax function can be found in B.6.

We use a similar method which is defined as Token Generation with Preferred Bit (TGPB) as in Algorithm 4 to generate a token. The algorithm TGPB takes prompt $\boldsymbol{\rho}$, previous output tokens $\boldsymbol{t}$, a preferred bit $b$ and tune factor $\tau$ as input. TGPB first employ an auto-regressive model $\mathsf{Model}(\cdot)$ to produce a vector of logits $\mathcal{D}$ of each word in the vocabulary $\mathcal{V}$. Let $\mathcal{D}[x]$ be the logits value of token $x$ in the vector. We use $\mathrm{LSB}(H(x))$ to denote the least significant bit of $H(x)$ for a token $x \in \mathcal{V}$. Let $\mathcal{S}_b$ be a subset of $\mathcal{V}$, a token $x \in \mathcal{S}_b$ if and only if $\mathrm{LSB}(H(x)) = b$. The $\mathcal{D}[x]$ is converted into a

normalized probability $p_x$ using the softmax function according to if it is in $\mathcal{S}_b$. The token $x$ with the highest probability $p_x$ will be returned.

---

**Algorithm 4** Token Generation with Preferred Bit (TGPB)

---

**Input:** $\boldsymbol{\rho}, \boldsymbol{t}, b, \tau$

```
/* The input bit b is preferred bit to be embed in the generated
token; the input τ is used to tune the probability that b will be
embedded correctly.  Note that, this algorithm is parameterized by the
vocabulary V and two disjoint subsets S₀ and S₁, where V  =  S₁ ∪ S₀ and
S₁ ∩ S₀  =  ∅.   Concretely, for b  ∈  {0,1}, a token x  ∈  S_b if and only if
LSB(H(x)) = b.   */
```

$\mathcal{D} \xleftarrow{\$} \mathsf{Model}(\boldsymbol{\rho}, \boldsymbol{t})$ //Run auto-regressive model and obtain the vector of logits.

$w \leftarrow 0$

**for all** $x \in \mathcal{V}$ **do**

   $\alpha_x \leftarrow \mathcal{D}[x]$ //Get the logits value of $x$.

   **if** $x \in \mathcal{S}_b$ **then**

      $w \leftarrow w + \exp\left(\alpha_x + \tau\right)$

   **else**

      $w \leftarrow w + \exp\left(\alpha_x\right)$

   **end if**

**end for**

**for all** $x \in \mathcal{V}$ **do**

   **if** $x \in \mathcal{S}_b$ **then**

      $p_x \leftarrow \frac{\exp\left(\alpha_x + \tau\right)}{w}$

   **else**

      $p_x \leftarrow \frac{\exp\left(\alpha_x\right)}{w}$

   **end if**

**end for**

$x \leftarrow \epsilon$

**for all** $y \in \mathcal{V}$ **do**

   **if** $p_y > p_x$ **then**

      $x \leftarrow y$

   **end if**

**end for**

return $x$

---

The input parameter $\tau$ is employed to modify the likelihood of selecting a token from the vocabulary. If a token $x \in \mathcal{S}_b$, its selection probability is heightened, and conversely, diminished otherwise. This approach skews the least significant hash value of the resulting token towards matching $b$. As the value of $\tau$ increases, the probability of the returned token $x$ satisfying $\mathrm{LSB}(H(x)) = b$ will rise. However, a larger $\tau$ value may disrupt the vocabulary distribution from the original output of $\mathsf{Model}(\boldsymbol{\rho}, \boldsymbol{t})$, potentially reducing the quality of the generated text.

It must be noticed that TGPB is a probabilistic watermark embedding algorithm. Whatever the value of $\tau$ is, the probability that $\mathrm{LSB}(H(x)) = b$ is less than 1. This means TGPB may generate a token that does not embed a bit of watermark correctly. We will show that the probability a bit $b$ is embedded correctly is high enough while the negligible impact on text quality is slight with suitable parameter $\tau$ in Section 6.

Both the short-range watermark and long-range watermark are embedded into tokens using TGPB algorithm. Note that the algorithm TGPB will introduce errors; These errors will be processed in two different ways:

**Short-range Watermark Error**    The short-range watermark is used to guarantee the robustness property. We treat the errors brought in TGPB the same as errors brought by the adversary. We use the edit distance to measure the similarity of the extracted watermark with the original one. If they are close enough we say the watermark is detected.

**Long-range Watermark Error**  The long-range watermark is used to guarantee the soundness property. Signature scheme is equipped to verify if an extracted watermark is the original one. The errors brought in TGPB must be corrected to recover the signature. The error correcting code is utilized to achieve this goal.

### E.1.3  GENERATIVE MODEL OF DUAL WATERMARKING

The Dual Watermarking of Generative Model (Watermark($\cdot$)) in Algorithm 5 is designed to generate watermarked text. Here, Watermark($\cdot$) takes private key sk and prompt $\boldsymbol{\rho}$ as input parameters. The expected output length is set as $n$.

---

**Algorithm 5** Dual Watermarking of Generative Model (Watermark)

---

**Input:** sk, $\boldsymbol{\rho}$
  $n \leftarrow$ target length
  $\boldsymbol{t} \leftarrow \epsilon, \pi_S \leftarrow \epsilon, \pi_L \leftarrow \epsilon$
  $IV \leftarrow$ "a constant string"
  $\tau \leftarrow c$
  **while** $|\boldsymbol{t}| < n$ **do**
    **if** $|\boldsymbol{t}| < n - \ell$ **then**
      **if** $|\pi_S| = 0$ **then**
        $\pi_S \leftarrow \mathsf{SWG}(IV)$
      **end if**
      $\bar{\sigma}_S \leftarrow \pi_S[0], \pi_S \leftarrow \pi_S[1:]$
      $x \leftarrow \mathsf{TGPB}(\boldsymbol{\rho}, \boldsymbol{t}, \bar{\sigma}_S, \tau)$
    **else**
      **if** $|\pi_L| = 0$ **then**
        $\pi_L \overset{\$}{\leftarrow} \mathsf{LWG}(\boldsymbol{t}, \mathsf{sk})$
      **end if**
      $\bar{\sigma}_L \leftarrow \pi_L[0], \pi_L \leftarrow \pi_L[1:]$
      $x \leftarrow \mathsf{TGPB}(\boldsymbol{\rho}, \boldsymbol{t}, \bar{\sigma}_L, \tau)$
    **end if**
    $\boldsymbol{t} \leftarrow \boldsymbol{t} \parallel x$
  **end while**
  return $\boldsymbol{t}$

---

The procedure that the tokens are generated with dual watermarks is illustrated in Figure 1.

**Generative Model of Short-range Watermark**  The short-range watermark is embedded periodically in every $m$ token except the last $\ell$ tokens. As the generation of the short-range watermark is from a constant initial vector, the short-range watermark remains the same in each period. The generative model generates the sequence of tokens which are embedded with the short-range watermark.

**Generative Model of Long-range Watermark**  The generation of the long-range watermark, on the other hand, depends on the tokens already generated which are embedded with the short-range watermark. The long-range watermark is only embedded once in the last $\ell$ tokens.

The SWG in Algorithm 2 and LWG in Algorithm 3 are used to generate short-range watermarks and long-range watermarks, respectively. The watermarks are embedded using the token generation with the preferred bit (TGPB) function in Algorithm 4. The factor $\tau \leftarrow c$ is used as a parameter to tune the probability that a watermark bit is correctly embedded in a token $x$.

The output tokens are generated one by one until the target length $n$ is reached. It should be noted that this algorithm does not guarantee that all the watermark bits are embedded correctly. As we mentioned in the Algorithm 4, some bits of the watermark may not be embedded correctly. This error should be tolerated in the detection algorithms.

E.2  DUAL WATERMARK DETECTOR

Dual watermark detector, Detect($\cdot$) also can be divided into two halves: Short-range Watermark Detector (SWD) in Algorithm 6 and Long-range Watermark Detector (LWD) in Algorithm 7.

**Short-range Watermark Detector**  In order to detect if a text $\boldsymbol{t}'$ contains the short-range watermark, all the substrings of $\boldsymbol{t}'$ will be checked. For one substring, each token is mapped to a bit using the hash function, thereby forming a bit string $\pi'_S$ of length $m$. Because the probabilistic watermark embedding Algorithm 4 is used, the extracted watermark may not be exactly the same as the original one. Then the edit distance between $\pi_S$ and $\pi'_S$, Distance($\pi_S, \pi'_S$), is used to measure if $\pi'_S$ is a valid watermark where $\pi_S$ is the hash value of the public initial vector $IV$. If Distance($\pi_S, \pi'_S$) is less than a predefined threshold $T$, then the output is true. If none of the substrings returns true then returns false.

---

**Algorithm 6** Short-range Watermark Detector (SWD)

---

**Input:** $\boldsymbol{t}', IV, T$
   $n \leftarrow |\boldsymbol{t}'|, i \leftarrow 0$
   **while** $i < n - (m + \ell)$ **do**
      $\pi_S \leftarrow H(IV)$
      $\pi'_S \leftarrow \epsilon, j \leftarrow 0$
      **while** $i + j < n - \ell$ **do**
         $\pi'_S \leftarrow \pi'_S \parallel \text{LSB}(H(\boldsymbol{t}'[i+j]))$
         $j \leftarrow j + 1$
         **if** Distance($\pi_S, \pi'_S$) $\leq T$ **then**
            return true
         **end if**
      **end while**
      $i \leftarrow i + 1$
   **end while**
   return false

---

**Long-range Watermark Detector**  The long-range watermark is embedded in the last $\ell$ tokens. Each of the last $\ell$ tokens is mapped to a bit using $\text{LSB}(H(\boldsymbol{t}[i]))$ and all the $\ell$ bits are composed into a bit string $\pi_L$. The $\pi_L$ is supposed to be the embedded watermark. The probabilistic embedding algorithm may bring errors into $\pi_L$ as discussed in Algorithm 4. ECC is used to recover the original signature $\sigma$ from $\pi_L$. The first $n - \ell$ tokens are used as the message to generate the signature $\sigma$ in Algorithm 5. If the input text is not modified, the signature verification will return true.

---

**Algorithm 7** Long-range Watermark Detector (LWD)

---

**Input:** $\boldsymbol{t}', \text{pk}$
   $n \leftarrow |\boldsymbol{t}'|, i \leftarrow 0, \texttt{plain} \leftarrow \epsilon, \pi_L \leftarrow \epsilon$
   **while** $i < n$ **do**
      **if** $i < n - \ell$ **then**
         $\texttt{plain} \leftarrow \texttt{plain} \parallel \boldsymbol{t}[i]$
      **else**
         $\pi_L \leftarrow \pi_L \parallel \text{LSB}(H(\boldsymbol{t}[i]))$
      **end if**
   **end while**
   $\sigma = \text{Decode}(\pi_L)$
   **if** Verify($\text{pk}, H(\texttt{plain}), \sigma$) $=$ true **then**
      return true
   **else**
      return false
   **end if**

---

We utilize both short-range watermark detector SWD and long-range watermark detector LWD in Detect($\cdot$) in Algorithm 8. Only when both watermarks are detected, it can be concluded that the text is generated by the generative model Watermark($\cdot$). When the short-range watermark is not detected, it can be inferred that the text is not generated by Watermark($\cdot$). If only the short-range watermark is detected, it can be inferred that the text is originally generated by Watermark($\cdot$) but has been tampered with. That is, if the return value $v_S = \mathtt{true}$, then it is a watermarked text; otherwise, it is not. If the return value $v_L = \mathtt{true}$, it is unmodified otherwise it is modified.

---

**Algorithm 8** Dual Watermark Detector (Detect)

**Input:** pk, $t'$
  {/* $T$ is a global parameter of threshold to detect short-range watermark.*/}
  $IV \leftarrow$ "a constant string"
  $\phi_r \leftarrow \mathsf{SWD}(t', IV, T)$
  $\phi_s \leftarrow \mathsf{LWD}(t', \mathsf{pk})$
  return $\langle \phi_r, \phi_s \rangle$

---

# F   Publicly-Detectable Dual Watermarking: Security Analysis

We will analyze the robustness property and soundness property of our publicly-detectable dual watermarking scheme PD2WS.

## F.1   Analysis of Watermark Errors

In Algorithm 4, a watermark bit $b$ is probabilistically embedded in a token $x$ by choosing $x$ such that $\mathsf{LSB}(H(x)) = b$. If a token $x$ satisfies that $\mathsf{LSB}(H(x)) = b$, we say $x$ is good otherwise it is bad. A good token means a bit of the watermark is embedded correctly and a bad token implies that an error bit of the watermark is embedded. We use $p_{\mathsf{good}}$ to denote the probability that a token $x$ is good and use $p_{\mathsf{bad}}$ to denote the probability that a token $x$ is bad.

$$p_{\mathsf{good}} = \Pr[\mathsf{LSB}(H(x)) = b],$$
$$p_{\mathsf{bad}} = \Pr[\mathsf{LSB}(H(x)) \neq b]. \tag{4}$$

It is obvious that $p_{\mathsf{good}} + p_{\mathsf{bad}} = 1$.

The probability $p_{\mathsf{good}}$ is adjusted by the factor $\tau$ using the softmax function. For a candidate token $x$, if $\mathsf{LSB}(H(x)) = b$, its probability of being chosen will increase according to the factor $\tau$. Otherwise, its probability of being chosen will decrease relatively. If we set $\tau = 0$ in Algorithm 4, the probability of tokens being chosen will not be tuned. In this case, we have the probability that $p_{\mathsf{good}} = \Pr[\mathsf{LSB}(H(x)) = b] = \frac{1}{2}$.

The probability that a token is good is independent of the other tokens. For any consecutive $n$ tokens that are generated in Algorithm 4, let $\alpha$ and $\beta$ be the number of tokens which are good and bad respectively. The expectation of $\alpha$ is $\mathbb{E}(\alpha) = n \cdot p_{\mathsf{good}}$ and the expectation of $\beta$ is $\mathbb{E}(\beta) = n \cdot p_{\mathsf{bad}}$.

Using the Chernoff bound as in Theorem B.5, we can measure the upper bound of $\beta$ with the following probability for any constant $\mu > 0$

$$\Pr[\beta \geq (1 + \mu)n \cdot p_{\mathsf{bad}}] \leq e^{-\Omega(n)}.$$

If $n = O(\lambda)$, we have

$$\Pr[\beta \geq (1 + \mu)n \cdot p_{\mathsf{bad}}] \leq \mathsf{negl}(\lambda). \tag{5}$$

## F.2   Security Proofs

We prove that our publicly-detectable dual watermarking scheme (PD2WS) can satisfy the completeness in Definition C.1, robustness in Definition 3.1, and soundness in Definition 3.2. We leave the distortion-freeness in Definition C.4 to be discussed in Section 6.

We recall the parameters that will be used in the following proofs. Let $m$ be the length of output of hash function $H(\cdot)$ where $m = O(\lambda)$. Let $\ell$ be the length of output of Encode($\cdot$) where $\ell = O(\lambda)$. Let $n = |t|$ be the length of text $t$ which is generated by Watermark($\cdot$). We assume $n \geq m + \ell$. Let $p_{\mathsf{bad}}$ be the probability that a generated token is bad as in the Equation 4.

### F.2.1 $\gamma$-COMPLETENESS

Our dual watermark algorithm uses two watermarking with different sensitivities to simultaneously ensure robustness and soundness.

Firstly, we will show short-range watermark detector will return `true` with overwhelming probability.

**Lemma F.1.** *Consider the publicly-detectable dual watermarking scheme* PD2WS = $(\mathsf{Setup}, \mathsf{Watermark}, \mathsf{Detect})$ *in Section 5 and assume that text $\boldsymbol{t}$ is generated by* $\mathsf{Watermark}(\cdot)$ *with the length $n \geq m + \ell$. Let $T$ be the threshold in Algorithm 6. If there exists a constant $\mu > 0$ such that $T \geq (1 + \mu) \cdot m \cdot p_{\mathsf{bad}}$, then we have* $\Pr[\mathsf{SWD}(\boldsymbol{t}, IV, T) = \mathtt{true}] \geq 1 - \mathsf{negl}(\lambda)$.

*Proof.* Let $\hat{\boldsymbol{t}}$ be the prefix string of $\boldsymbol{t}$ with $m$ tokens. For $n \geq m + \ell$, the short-range watermark $\pi_S$ must be embedded in $\hat{\boldsymbol{t}}$ (with errors). Let $\beta$ be the number of bad tokens in $\hat{\boldsymbol{t}}$. From the Equation 5, we have $\Pr[\beta \geq (1+\mu)m \cdot p_{\mathsf{bad}}] \leq \mathsf{negl}(\lambda)$. For $T \geq (1+\mu)m \cdot p_{\mathsf{bad}}$, we have $\Pr[T \geq \beta] \geq 1 - \mathsf{negl}(\lambda)$.

Let $\pi'_S$ be the watermark extracted in SWD. We have the distance between $\pi_S$ and $\pi'_S$ as $\mathsf{Distance}(\pi_S, \pi'_S) = \beta$. If $T \geq \beta$, $\mathsf{SWD}(\boldsymbol{t}, IV, T)$ will return `true`. Putting them together, we have $\Pr[\mathsf{SWD}(\boldsymbol{t}, IV, T) = \mathtt{true}] \geq 1 - \mathsf{negl}(\lambda)$. $\qquad\square$

Secondly, we will show long-range watermark detector will also return `true` with overwhelming probability.

**Lemma F.2.** *Consider the publicly-detectable dual watermarking scheme* PD2WS = $(\mathsf{Setup}, \mathsf{Watermark}, \mathsf{Detect})$ *in Section 5, and assume that text $\boldsymbol{t}$ is generated by* $\mathsf{Watermark}(\cdot)$ *with the length $n \geq m + \ell$. Let $d$ be the number of errors that* $\mathsf{Decode}()$ *can correct in Algorithm 7. Assume that the signature scheme $\Sigma$ is complete in Algorithm 7. If there exists a constant $\mu > 0$ such that $d \geq (1 + \mu) \cdot \ell \cdot p_{\mathsf{bad}}$, then we have* $\Pr[\mathsf{LWD}(\boldsymbol{t}, \mathsf{pk}) = \mathtt{true}] \geq 1 - \mathsf{negl}(\lambda)$.

*Proof.* Let $\hat{\boldsymbol{t}}$ be the last $\ell$ tokens of $\boldsymbol{t}$. For $n \geq m + \ell$, the long-range watermark $\pi_L$ must be embedded in $\hat{\boldsymbol{t}}$ (with errors). Let $\beta$ be the number of bad tokens in $\hat{\boldsymbol{t}}$. From the Equation 5, we have $\Pr[\beta > (1+\mu)\ell \cdot p_{\mathsf{bad}}] \leq \mathsf{negl}(\lambda)$. For $d \geq (1+\mu)\ell \cdot p_{\mathsf{bad}}$, we have $\Pr[d \geq \beta] \geq 1 - \mathsf{negl}(\lambda)$. Because ECC can correct $d$ errors, if $d \geq \beta$ then $\sigma = \mathsf{Decode}(\pi_L)$ in Algorithm 7. For the signature scheme is complete, given a correct signature $\sigma$, $\Pr[\mathsf{Verify}(\mathsf{pk}, H(\mathtt{plain}), \sigma) = \mathtt{true}] \geq 1 - \mathsf{negl}(\lambda)$. If the $\mathsf{Verify}(\cdot) = \mathtt{true}$ then $\mathsf{LWD}(\cdot)$ will return `true`. Putting them together, we have $\Pr[\mathsf{LWD}(\boldsymbol{t}, \mathsf{pk}) = \mathtt{true}] \geq (1 - \mathsf{negl}(\lambda))^2 = 1 - \mathsf{negl}(\lambda)$. $\qquad\square$

Based on Lemma F.1 and Lemma F.2, we can prove the completeness property.

**Theorem F.3** ($\gamma$-Completeness)**.** *Consider the publicly-detectable dual watermarking scheme* PD2WS = $(\mathsf{Setup}, \mathsf{Watermark}, \mathsf{Detect})$ *in Section 5 with the same parameters as in Lemma F.1 and in Lemma F.2. Let $\gamma = m + \ell$. We have that* PD2WS *is $\gamma$-complete.*

*Proof.* Let text $\boldsymbol{t}$ is generated by $\mathsf{Watermark}(\cdot)$ and $|\boldsymbol{t}| \geq \gamma$. From Lemma F.1 we have $\Pr[\mathsf{SWD}(\boldsymbol{t}, IV, T) = \mathtt{true}] \geq 1 - \mathsf{negl}(\lambda)$. From Lemma F.2 we have $\Pr[\mathsf{LWD}(\boldsymbol{t}, \mathsf{pk}) = \mathtt{true}] \geq 1 - \mathsf{negl}(\lambda)$. Let $\langle \phi_r, \phi_s \rangle = \mathsf{Detect}(\cdot)$ as in Algorithm 8. We have

$$\Pr[(\phi_r = \mathtt{false} \vee \phi_s = \mathtt{false}) \wedge (|\boldsymbol{t}| \geq \gamma)]$$
$$\leq \Pr[\mathsf{SWD}(\boldsymbol{t}, IV, T) = \mathtt{false}] + \Pr[\mathsf{LWD}(\boldsymbol{t}, \mathsf{pk}) = \mathtt{false}]$$
$$\leq \mathsf{negl}(\lambda).$$

$\qquad\square$

### F.2.2 d-ROBUSTNESS

The short-range watermark based on edit distance is not sensitive to token modifications, thus it can verify the watermark as true for slightly modified text, ensuring robustness. We prove **d**-Robustness using short-range watermark.

First, we will demonstrate that if the distance between two texts is bounded by a parameter $n$, then there exist two corresponding substrings of the texts where the distance is bounded by $\frac{n}{m}$ when the text is divided into $m$ substrings.

**Theorem F.4.** *Let $t'$ and $t$ be two texts, the distance of the two texts is $\mathsf{Distance}(t', t) = n$. Assume that $t$ is divided into $m$ consecutive substrings $\hat{t}_i$ where $i \in \{1, m\}$ as $t = \hat{t}_1, \cdots, \hat{t}_m$. There is a substring $\hat{t}'_i$ of $t'$ and a substring $\hat{t}_i$ of $t$ such that $\mathsf{Distance}(\hat{t}'_i, \hat{t}_i) \leq \frac{n}{m}$.*

*Proof.* For each substring $\hat{t}_i$ where $i \in \{1, m\}$ choose the substring $\hat{t}'_i$ of $t'$ with the least distance $\mathsf{Distance}(\hat{t}'_i, \hat{t}_i)$, we have that $\mathsf{Distance}(t', t) \geq \sum_{i=1}^{m} \mathsf{Distance}(\hat{t}'_i, \hat{t}_i)$. If for all $\hat{t}'_i$ and $\hat{t}_i$ it is that $\mathsf{Distance}(\hat{t}'_i, \hat{t}_i) > \frac{n}{m}$, then we have $\mathsf{Distance}(t', t) > m \cdot \frac{n}{m} = n$. It is contradicted with the condition that $\mathsf{Distance}(t', t) = n$. □

Now, we can prove **d**-Robustness property.

**Theorem F.5** (**d**-Robustness). *Consider the publicly-detectable dual watermarking scheme* PD2WS $=$ (Setup, Watermark, Detect) *in Section 5, and assume that text $t$ is generated by* Watermark$(\cdot)$ *with the length $n \geq m + \ell$. Let $T$ be the threshold in Algorithm 6. If there exists a constant $\mu > 0$ such that $T \geq (1 + \mu) \cdot m \cdot p_{\mathsf{bad}} + \frac{m}{n-\ell}d$, then we have that* PD2WS *is* **d**-*robust.*

*Proof.* Let $t' \leftarrow \mathcal{A}(t)$ and the edit distance between $t'$ and $t$ is $\mathbf{d} = \mathsf{Distance}(t, t')$. The text $t$ is divided into $\frac{n-\ell}{m}$ segments to embed short-range watermark in Watermark. With Theorem F.4, for the $\mathbf{d} = \mathsf{Distance}(t, t')$, there is at least one substring $\hat{t}'$ in $t'$ and corresponding substring $\hat{t}$ in $t$ that $\mathsf{Distance}(\hat{t}, \hat{t}') \leq \frac{m}{n-\ell}\mathbf{d}$.

Similar with the proof of Lemma F.1, let $\beta$ be the number of bad tokens in $\hat{t}$, we have $\Pr[\beta \geq (1 + \mu)m \cdot p_{\mathsf{bad}}] \leq \mathsf{negl}(\lambda)$. Let $\beta'$ be the number of bad tokens in $\hat{t}'$, we have $\beta' \leq \beta + \frac{m}{n-\ell}\mathbf{d}$. That is $\Pr[\beta' \geq (1 + \mu)m \cdot p_{\mathsf{bad}} + \frac{m}{n-\ell}\mathbf{d}] \leq \mathsf{negl}(\lambda)$.

For $T \geq (1 + \mu)m \cdot p_{\mathsf{bad}} + \frac{m}{n-\ell}\mathbf{d}$, we have $\Pr[T \geq \beta'] \geq 1 - \mathsf{negl}(\lambda)$. If $T \geq \beta'$, $\mathsf{SWD}(t', IV, T)$ will return true.

Putting them together, we have $\Pr[\mathsf{SWD}(t', IV, T) = \mathtt{true}] \geq 1 - \mathsf{negl}(\lambda)$. That is, the SWD will return $\phi_r = \mathtt{true}$ with probability $\Pr[\phi_r = \mathtt{true}] \geq 1 - \mathsf{negl}(\lambda)$. Let $\langle \phi_r, \phi_s \rangle = \mathsf{Detect}(\cdot)$ as in Algorithm 8. We have $\Pr[\phi_r = \mathtt{false}] < \mathsf{negl}(\lambda)$. □

### F.2.3 D-SOUNDNESS

On the other hand, the long-range watermark based on digital signatures is very sensitive to token modifications, and it verifies the watermark as false for changed text, ensuring soundness. We prove **d**-Soundness with long-range watermark.

**Theorem F.6** (**d**-Soundness). *Consider the publicly-detectable dual watermarking scheme* PD2WS $=$ (Setup, Watermark, Detect) *in Section 5, assume that the signature scheme $\Sigma$ is unforgeable in Algorithm 7. If $\mathbf{d} > \ell$, then we have that* PD2WS *is* **d**-*sound.*

*Proof.* The adversary queries the oracle Watermark$(\cdot)$ and get a text set $\mathcal{Q}$ and then generate a text $t' \leftarrow \mathcal{A}(\mathsf{pk})$ satisfying the condition $\mathsf{Distance}(t', \mathcal{Q}) \geq \mathbf{d}$. Comparing $t'$ with any $t \in \mathcal{Q}$, because $\mathbf{d} > \ell$ there is at least one token which is different in $t'$ and $t$ before the last $\ell$ tokens. That is the message `plain` verified in SWD is different from any one signed in LWG in the querying stage.

Given the signature $\Sigma$ scheme is unforgeable, the probability that the signature verification return true is negligible. That is the SWD will return $\phi_s = \mathtt{true}$ with probability $\Pr[\phi_s = \mathtt{true}] \leq \mathsf{negl}(\lambda)$. Let $\langle \phi_r, \phi_s \rangle = \mathsf{Detect}(\cdot)$ as in Algorithm 8. We have $\Pr[\phi_s = \mathtt{true}] < \mathsf{negl}(\lambda)$. □

### F.2.4 COMBINE D-ROBUSTNESS AND D-SOUNDNESS

We will show that with proper parameters, the **d**-Robustness and **d**-Soundness can be achieved simultaneously.

**Theorem F.7.** *Consider the publicly-detectable dual watermarking scheme* PD2WS $=$ (Setup, Watermark, Detect) *in Section 5, following all the parameters in Theorem F.5 and F.6.*

*If the parameters satisfy that $\frac{n-\ell}{\ell}(\frac{T}{m} - (1 + \mu) \cdot p_{\mathsf{bad}}) > 1$, then we have that* PD2WS *is* **d**-*robust and* **d**-*sound, simultaneously.*

*Proof.* Let $\mathbf{d} = \frac{n-\ell}{m}T - (1 + \mu)(n - \ell) \cdot p_{\mathsf{bad}}$, we have

$$T = (1 + \mu)m \cdot p_{\mathsf{bad}} + \frac{m}{n - \ell}\mathbf{d},$$

which satisfy the condition in Theorem F.5. That is PD2WS is ***d-robust***.

Considering the condition that $\frac{n-\ell}{\ell}(\frac{T}{m} - (1 + \mu) \cdot p_{\mathsf{bad}}) > 1$, we have $\mathbf{d} > \ell$ which satisfy the condition in Theorem F.6. That is PD2WS is ***d-sound***. □

Let $\theta = \frac{T}{m} - (1 + \mu) \cdot p_{\mathsf{bad}}$. If $p_{\mathsf{bad}}$ is small enough, we can choose suitable $T$ and $m$ such that $\theta > 0$. Under this condition, we obtain that $\mathbf{d} = (n - \ell) \cdot \theta$. We will show in Evaluation 6 that small $p_{\mathsf{bad}}$ is achievable.

# G    ADDITIONAL EXPERIMENT RESULTS

Table 1 shows the completion situations of randomly selected prompts from C4 dataset Raffel et al. (2020) without watermark (NW) and with watermark (W) under different tune factor $\tau$ using OPT-2.7B. We witness that the PPL values of the watermarked completion slowly increase with the increase of the tune factor, which shows that the text quality has declined to a certain extent. However, the actual generated text has no quality defects visible to humans.

Table 2 provides another example illustrating how the output text changes as $\tau$ increases, given the same prompt.

| Prompt | Real Completion | Non-Watermarked (NW) | Watermarked (W) | PPL(NW) | PPL(W) | $\tau$ |
|---|---|---|---|---|---|---|
| [...] Shortly after the release of the lead single "Life's About to Get Good," it hit number one on the Billboard 200. Of course, fans were simply dying to see how their favourite country icon was doing, so the new single rocked its way to the top of Billboard + Twitter Trending 140, too! So far, the song | 's video has over four million views on YouTube and the number continues to ride the wave of Now's release. The 52-year-old country star, still clad in her iconic leopard print, insists that Now is not a break up album but another coming-of-age collection of tunes [...truncated] | has racked up over 1.5 million streams on Spotify. The album is also set to be certified platinum in Canada, which means it's already sold over 100,000 copies. "I'm so excited to share this album with you," Shania wrote on Twitter. [...truncated] | has racked up more than 1.2 million streams on Spotify. The album also features the likes of "The Man" and "The Woman," which were both released in June. "I'm so excited to share this album with you. I hope you love it as much as I do," Shania wrote on Instagram. [...truncated] | 1.65 | 1.86 | 1 |
| [...] Health authorities in New Zealand said that about 200 passengers on the Dawn Princess ship became infected with the norovirus. The ship was scheduled to leave for Australia on Monday. The last time there was a norovirus | outbreak on the ship was back in 2012. According to Yahoo, health officials conducted a series of tests, and they confirmed that the illness was norovirus, but the outbreak does seem to be going away. [...truncated] | outbreak on a cruise ship was in 2003, when about 1,000 passengers and crew were infected. The ship was carrying about 1,000 passengers and crew from New Zealand, Australia, the United States and Canada. [...truncated] | outbreak was on a cruise ship in Australia in 2014. New Zealand's Ministry of Health said that about 200 people on board were infected with norovirus. "The passengers are being treated in isolation and the ship will be held [...truncated] | 1.59 | 1.91 | 2 |
| [...] automotive datasets from IHS Markit that include vehicle sales and production history, aggregated registration information, manufacturing volumes and vehicle pricing insight for multiple geographies. According to Greenwich Associates, 50 per cent of institutional | investors are planning to increase their usage of alternative datasets. Research Signals delivers valuable investment insights through a comprehensive library of more than 600 global stock [...truncated] | investors are now using alternative data to make investment decisions. "The launch of our new stock selection and strategy signals for the automotive sector is a significant step forward in our efforts to provide our clients with the most comprehensive [...truncated] | asset managers have invested in automotive stocks, with more than $2 trillion of assets under management. "The launch of these new indicators for the automotive sector represents another significant step for the Research Signals service, which is [...truncated] | 1.28 | 2.21 | 3 |
| [...] Assembled by Palestinian artist Said Baalbaki, the exhibition presents 50 of Abbo's sketches, etchings and object, as well as texts – all taken from Baalbaki's personal collection of work from, and publications on, the elusive sculptor. Baalbaki has | no work of his own in this show. Born in Safad, northern Palestine, around 1888-1890, Abbo was a farmer and fisherman who later took up carpentry and stone masonry. It was not until he moved to Berlin in 1911 to study sculpture [...truncated] | been a longtime friend of Abbo's, and has been working with him on the exhibition for the past two years. "I have been working with him for a long time, and I have been collecting his work for a long time," Baalbaki said. [...truncated] | curated the show, which runs until August 31, with support from the Palestinian Cultural Fund. "I wanted this exhibition because I wanted it to be a celebration, a celebration for the Palestinian artist," [...truncated] | 1.68 | 2.51 | 4 |

Table 1: Example text completions by OPT-2.7B Zhang et al. (2022) with different tune factor $\tau$. Prompts are randomly selected from the news-like subset of the C4 dataset Raffel et al. (2020).

| Prompt | Real Completion | Watermarked Output | PPL | $\tau$ |
|---|---|---|---|---|
| [...] Walmart is big. The acquisitions of Jet.com, Bonobos, ModCloth, etc. have only made it bigger and more complex, saying nothing too of the ever-changing demands of consumers and Walmart's ever-expanding interests abroad (see Flipkart). Therefore, it is only right that someone, like Casey, take up the mantle to oversee and to fight for the consumer and to ensure that all Walmart's activities look, feel, and convey the Walmart brand in the simplest, most straightforward way as possible. The products inside Walmart's stores are immaterial to its future success. What matters is the Product of its brand—its website, its store, its app-based services, etc. Those are the Products that matter. In a future world where the only thing that differentiates a physical from a digital experience is the memory and | delight of being somewhere, a Walmart store is the Product or collective set of experiences that will get someone off his or her couch. A tube of toothpaste just won't cut it anymore. While the products within Walmart's store or on its website will come and go and ebb and flow, the shroud of the Walmart brand will be what matters. [...truncated] | the brand experience, Walmart's new hire is a must have and a must have now for the future of Walmart. Walmart has hired the world's most famous Product Manager, Jeff Bezos, to lead its new Design and Product team. This month some important Walmart news did not get the attention it deserved. The news wasn't some shiny announcement about virtual reality or about [...truncated] | 1.49 | 0 |
| | | the experience, Walmart's new head of design will be the one who will make sure that all Walmart's products and services, from websites to employee and consumer apps, work in cohesion and from a singular experience design point of view. Walmart's new head of design will be the one who will make sure that all Walmart's products [...truncated] | 1.57 | 1 |
| | | the brand, Walmart's new head of design will be the one who will make sure that all Walmart's products and services, both digital and physical, work together and from a singular experience design point of view. Walmart's new head of design will be the one who will make sure that all Walmart's products and services [...truncated] | 2.41 | 2 |
| | | the brand experience, Walmart's new hire is a must have. Walmart has hired the world's most famous Product Manager, Jeff Bezos, to lead its new Design and Product team. This month some important Walmart news did not get the attention it deserved. The news wasn't some flashy announcement about virtual reality or about some [...truncated] | 2.57 | 3 |
| | | the brand experience, Walmart's new hire is a must have and a must have now for the future of Walmart. Walmart has hired the world's most famous Product Manager, Jeff Bezos, to lead its new Design and Product team. This month some important Walmart news did not get the attention [...truncated] | 2.82 | 4 |

Table 2: Example text completions by OPT-2.7B Zhang et al. (2022) with different tune factor $\tau$ and same prompt.