# OpenReview forum: "Two Halves Make a Whole: How to Reconcile Soundness and Robustness in Watermarking for Large Language Models"
_ICLR.cc/2025/Conference — Submitted to ICLR 2025_

### Official Review · Reviewer_UkDR · 2024-10-21

**Soundness:** 1
**Presentation:** 2
**Contribution:** 2
**Rating:** 3
**Confidence:** 5

**Summary:**

The paper proposes embedding two watermarks: one for robustness (high detectability even under attack) and one for integrity (detection of modifications).

**Strengths:**

**S1. It is a priori a good idea to jointly design a robust watermark and a fragile watermarking schemes.**

**S2. It is a good idea to point the flaws of the paper Fairoze et al. (2023).**

I wish it could have been more severely outlined.

**Weaknesses:**

**W1. Lack of a critical point of view of paper Fairoze et al. (2023).**

Paper Fairoze et al. (2023) has not yet been published, and I can anticipate why. The authors present their work as a robust watermarking scheme, but it is not at all robust. It is useless for GenIA detection. Indeed, this is a fragile watermarking scheme that targets tampering detection, ie. protection of the integrity of the generated text.

This paper does not question the confusion brought by Fairoze et al. (2023).... or too politely.

**W2. Confusion in the definitions of completeness and soundness**

I disagree with line 41 that robustness and soundness have been defined in  Christ et al. (2023); Fairoze et al. (2023).
This is true, but these are not the same definitions! This makes sense because Christ studies robust watermarking whereas Fairoze targets fragile watermarking (even if they do not admit it).

Since the submitted paper proposes a double watermarking schemes, each of them should have its own definitions of completeness and soundness:
* Robust watermarking (called short-range watermark in the proposed scheme) should adhere to the definitions of Christ et al.:
Soundness holds if false positive rate is under control; Completeness holds if generated text is detectable if enough entropy.
* Fragile watermarking (called long-range watermark in the proposed scheme) should adhere to the definitions of Fairoze et al.:
Soudness holds if it is impossible to forge a signed text without the knowledge of the private key; Completeness holds if it possible to embed the signature almost surely (provided enough entropy).

Yet, this paper brings confusion by taking the soundness of one and the completeness of the other.

**W3. Soundness and edit distance**

This is especially true for the soundness of the robust watermark. No guarantee is given concerning the probability of a false positive.
And I do not know how it would be possible due to using the edit distance. For instance, Kuditipudi et al. only empirically measure this probability with a long Monte Carlo simulation. Moreover, the authors propose to compute the edit distance over all substrings -> this will lead to a very high false positive rate.

**W4. Public-key detection**

I am highly doubtful about the public-key detection scheme.
- For fragile watermarking: this was done in traditional watermarking literature a long time ago. I am ok with this.
- For robust watermarking: this does not make sense. The attacker knows where the short-range watermarks are embedded and can concentrate his attacks on those chunks. Revealing the way you detect obviously weakens the robustness of the watermark.

**W5. Technical details**
The technical description of the proposed scheme is quickly packed at the end of the paper. It is very difficult to understand the full picture; one has to delve into the appendices to grab the details.

**Questions:**

Q1. Appendix F.1. What is \mu?

Q2. What is the use of hash function H? Especially H(IV)? Why not directly embedding IV?

Q3. How does the LLM know that it is about to generate the last $\ell$ tokens?

Q4. Examples shown in Tables 1 & 2 are misleading as they are truncated. It does not reveal the fact that the schemes produce very very long texts (more than 2000 tokens), whereas known robust watermarking techniques in the literature work with ~100 tokens.

---

> ### Author Response · Authors · 2024-12-03
>
> ## Regarding "W4. Public-key detection"
>
> In our construction, the watermarked information has been embed in all tokens; using the public Detect algorithm, it may be possible to remove the watermarked information in some of the tokens (e.g., by altering those tokens).
> When a significant fraction of tokens in the text is modified, the watermark in the maliciously altered version becomes undetectable. However, this heavily modified version loses its utility, as the substantial alterations render the text ineffective or meaningless.
>
>
> In our paper, the attacker is not allowed to modify the text significantly since the attacker is expecting to produce a useful version.
>
>
> We further remark that, even in the context of private watermarking, the attacker by altering all tokens, can produce a useless version trivially.
>
> Please also see our explanation in the introduction of our submission, lines 50-57,
> >the robustness property requires that even if a watermarked text has been modified, the embedded watermark should not be eliminated; instead, it should still be able to be detected.
>
> >the adversary is not allowed to remove the entire watermarked text; instead, the modified version from the adversary, denoted as $\mathbf{t}'$, and the original version of the watermarked text, denoted as $\mathbf{t}$, must have an overlapping of at least a $\delta$-length segment, where $\delta\in \mathbb{N}$.

---

### Official Review · Reviewer_hG1k · 2024-10-22

**Soundness:** 3
**Presentation:** 2
**Contribution:** 2
**Rating:** 5
**Confidence:** 3

**Summary:**

The paper deals with the properties of publicly detectable watermarks (introduced as in [1]). In particular, watermarks are embedded via rejection sampling, embedding a bitstring across the individual tokens (via LSB of a token hash). In some cases, it additionally employs ECCs to deal with the inherent randomness in the embedding procedure. The authors first show that (using the soundness and robustness definitions of [1]) such a watermark can not achieve soundness and robustness simultaneously under the same parameter $d$. The paper then proposes a remedy to this by combining two versions of the watermark idea in a "dual" watermark where a long-range watermark provides the d-soundness and a short-range watermark provides the d-robustness.

**Strengths:**

- The analysis of impossibility results seems reasonable and correct to me. Additionally, the proposed PD2WS technically addresses this (see below for some caveats I have on this).
- Moving from substring adversaries to editing adversaries is definitely a step forward in terms of adversarial setup.
- The idea of PD2WS is very straightforward as it combines the robustness short range watermark with a signature that is encoded via the LWG (as you would do a signature in cryptography).

**Weaknesses:**

- The experimental evaluation is very limited. The section explains a decent amount of background (that, while appreciated, should not be needed for the audience at ICLR) but only presents small experiments for individual parts of the construction.
    - Text quality is evaluated only on PPL (where, for Mistral, we can see noticeable degradation) and only on **20** queries from C4. Qualitative examples are only from OPT-2.7B, which is quite outdated and shows noticeably less decrease in quality than more realistic models (Figure 4).
    - There are no other further experiments ablating practical robustness (6.3 deals with this in a theoretical nature), nor, e.g., potential results in case a more realistic adversary (commonly paraphrasing adversaries are considered in LLM watermarking literature) that could help underline the practical robustness of PD2WS.
    - Similarly, there is no note on runtime as in [1].
    - The example of 6.3 assumes a minimum length of m+l > 512+256 tokens. Such requirements would severely limit the practicality of a watermark. It would make sense to better ablate these choices, w.r.t., robustness/applicability.
- The main method of the paper (PD2WS) is only properly explained on 4 pages in Appendix E. The corresponding Figure 1 was hard to parse without having read the Appendix with "seemingly random" (making sense after the Appendix) introductions of, e.g., an IV.

### Typos/Nits
- The paper claims that it is the "first to study a more realistic editing adversary" (L104-105). However, it is only the first to do it for this class of watermarks. In particular, such adversaries have already been studied in [2,3] for more popular watermarks.
- L14 LLMs
- L115 "an" editing adversary
- L136 Citet used where citep would fit

[1] Fairoze, Jaiden, et al. "Publicly detectable watermarking for language models." _arXiv preprint arXiv:2310.18491_ (2023).

[2] Kuditipudi, Rohith, et al. "Robust distortion-free watermarks for language models." _arXiv preprint arXiv:2307.15593_ (2023).

[3] Kirchenbauer, John, et al. "A watermark for large language models." _International Conference on Machine Learning_. PMLR, 2023.

**Questions:**

- Can you provide some additional experimental evaluations and adress the length concerns as specified above?

---

### Official Review · Reviewer_Bvw7 · 2024-11-03

**Soundness:** 3
**Presentation:** 3
**Contribution:** 2
**Rating:** 6
**Confidence:** 3

**Summary:**

This paper establishes an impossibility result, showing that achieving both robustness and soundness properties is unattainable through a publicly-detectable single watermarking scheme. Additionally, it introduces a feasible solution by pioneering the concept of a publicly-detectable dual watermarking scheme for AI-generated content.

**Strengths:**

The paper is well explained and supported with sufficient math notations.
The new concept of publicly-detectable dual watermarking scheme could contribute to the advancement of LLM watermarking in terms of safety, privacy.

**Weaknesses:**

The chapters before introducing the proposed method are a bit too lengthy.

The paper could include more analysis of the proposed method in terms of other important properties of LLM watermarking.

**Questions:**

As explained in Weaknesses.

---

### Official Review · Reviewer_xvqD · 2024-11-04

**Soundness:** 2
**Presentation:** 3
**Contribution:** 1
**Rating:** 3
**Confidence:** 4

**Summary:**

The paper is based on the following two observations:
1. If you want your watermark to be unforgeable ("sound"), then it better not be robust. If it is robust, then anyone can simply maul a watermarked response, obtaining a new response that will still have the watermark --- violating unforgeability.
2. It is still possible to get the best of both worlds, by introducing two separate detection keys: One that is robust, and the other that is unforgeable.
They then present such a scheme achieving the best of both worlds, and perform experiments with this scheme.

**Strengths:**

Watermarks with unforgeable public detectors could be a useful tool for attributing harmful generations to bad models.
Therefore, it is valuable to point out that the scheme of Fairoze et al. is attempting to achieve contradictory goals.
It is also a useful contribution to construct a scheme that resolves the contradiction, and to perform experiments with it.

**Weaknesses:**

Certain aspects of the paper, which are emphasized as being novel, are not in fact new to this work. Given the large number of papers about watermarking, it is extremely reasonable that the authors missed these, but it nonetheless undermines the claims of novelty:
- I'm aware of at least one other work that also recognized and addressed the dilemma between unforgeability and robustness: "Pseudorandom error-correcting codes," Section 2.6.
- In the same paper, they also consider adversaries that make random deletions. The paper "Edit Distance Robust Watermarks for Language Models" also considers adversaries that make a broader class of edits.

Neither of the above papers contain experiments, so this paper could have some claim to novelty in this regard. However, editing adversaries are widely considered in basically every paper that does watermarking robustness experiments --- for instance, "Provable Robust Watermarking for AI-Generated Text" even has a robustness experiment that they call the "Editing attack."

In any case, the experiments leave much to be desired:
- They do not compare the quality-robustness trade-off to existing schemes at all.
- They only use perplexity to measure quality, which is known to prefer repetitive text and therefore may be favorable to watermarks. There is work on more useful quality evaluations for watermarks, such as the paper "MARKMYWORDS: Analyzing and Evaluating Language Model Watermarks." They do not theoretically analyze their quality, either, and I suspect that they would only achieve the degenerate 1-distortion freeness under their definition if they did. (This is because their definition of distortion-freeness allows the distinguisher to make many queries to the model before making their decision, as in the definitions of Christ et al. and Fairoze et al.)
- There do not appear to be any experiments demonstrating the actual detectability of their watermark. And their calculations suggest using 1000s of tokens, which is extremely inefficient.

Finally, a minor comment: I found the discussion around public detectability misleading. It is not the case that schemes aside from Fairoze et al. are inherently "secret-key," as one can always simply publish the key and achieve a fully functional scheme. The difference is in the *unforgeability* despite the fact that the detection key is made public.

**Questions:**

- In the appendix, you include several results on things like completeness and soundness. However, these are difficult to parse due to the presence of all these parameters. For instance, it would be much nicer if Lemma F.1 simply said something like "our scheme is \gamma-complete for any text with [some notion of entropy] at least H."
- Is there a reason that you don't simply take any existing watermarking scheme that can encode messages, and use that to encode a signature of the previously-output text? In particular, it seems unnecessary to come up with an entirely new watermarking scheme of your own; in the end, it seems that your ideas would work identically using any underlying multi-bit watermark as a black box.

---

> ### Author Response · Authors · 2024-12-03
>
> Thank a lot for your valuable comments and suggestions!
>
> First, we want to clarify that our paper has been submitted to USENIX Security '24 Winter (with submission deadline: February 8, 2024), and then got rejected; in this submission, the implementation and evaluation part has been updated by including those based on LLaMA-7B, while the theory part (including definitions, the impossibility proof, construction and security proofs) remain the same.
> We were not aware that the work by Christ and Gunn (Crypto 2024), "Pseudorandom error-correcting codes," is very related to ours, and did not cite this important result in our paper.
>
> ---
>
> ## Comparison to the result by Christ and Gunn (Crypto 2024)
> After reading the paper by Christ and Gunn (Crypto 2024; https://arxiv.org/abs/2402.09370), we have the following.
>
> As you pointed out that, in section 2.6 of the paper by Christ and Gunn (Crypto 2024), the dilemma between unforgeability and robustness has been discussed. However, our discovery is **independent** of theirs. More importantly, **we provide a formal impossibility proof** in our submission, while they only had informal discussions.
>
> In addition, among many other results, Christ and Gunn (Crypto 2024) provide a construction (that related to ours) in section 7.4 Watermarks with public attribution;
> This construction can achieve undetectablility, unforgeability; however, it is unclear to us if this construction can achieve all three security properties undetectablility, unforgeability and robustness. Note that, as already pointed out by Chris and Gunn in their paper, page 56 beginning paragraph, quoted as follows
>
> >Although AttrText is intentionally not robust, our scheme’s Detect function retains all properties (undetectability, robustness, soundness) of our standard watermarks.
>
> In Detect function (see Algorithm 3, page 50), our understanding is that sk  i.e., (PRC.sk, a) must be hidden; otherwise, undetectablility cannot be achieved. In AttrText function (see Algorithm 6, page 57), our understanding is that (PRC.sk, a) is public since (PRC.sk, a)  is already part of the pk. *When (PRC.sk, a) is public, indeed, the unforgeability property can be achieved; unfortunately now undetectablility cannot be achieved.* In this sense, we believe their theorem 8 in section 7.4 should be modified and make it explicit that **their construction cannot achieve undetectablility, unforgeability and robustness, at the same time**.
>
> In our construction, the secret key is only used in Watermark generation algorithm, and there is no secret key in our Detect algorithm; therefore **our construction can achieve unforgeability and robustness at the same time**.
>
>
>
>
> Finally, since our goal is not for achieving undetectablility, we do not require the LPN assumption. We can achieve practical, publicly detectable watermarking using only MiniCrypt assumptions (while the construction by Chris and Gunn cannot).
>
>
>
> ---
>
> ## Regarding your suggestions/comments about "blackbox construction":
>
> > Is there a reason that you don't simply take any existing watermarking scheme that can encode messages, and use that to encode a signature of the previously-output text? In particular, it seems unnecessary to come up with an entirely new watermarking scheme of your own; in the end, it seems that your ideas would work identically using any underlying multi-bit watermark as a black box.
>
>
>
> Thanks a lot for your suggestion. As mentioned, our early submission was sent to USENIX security, Feb 8th, 2024. We were not aware of any suitable watermarking schemes for our purpose here (i.e., for constructing a publicly detectable watermarking scheme). We will consider your suggestions in our future version.
> In addition, based on our knowledge, we believe we are the first to use edit distance to detect watermarks in a watermarking scheme.

---

### Meta-Review · Area_Chair_EPds · 2024-12-18

**Metareview:**

Summary: This paper presents an impossibility result, demonstrating that a publicly-detectable single watermarking scheme cannot simultaneously achieve both robustness and soundness. It also proposes a practical solution by introducing the novel concept of a publicly-detectable dual watermarking scheme for AI-generated content, by enabling two keys: one for robustness while the other for soundness.

Strengths:
1. Watermarking with both good robustness and soundness is an important research topic. The paper proposes a method PD2WS to technically address this trade-off.

Weaknesses:
1. Reviewers have concerns on the novelty of this paper. Many concepts such as 1) the dilemma between unforgeability and robustness and 2) adversaries that make a broader class of edits are not new to the reviewers. This weakness is not a big deal as Christ and Gunn (Crypto 2024 in Aug 2024) is regarded as a concurrent work according to ICLR guideline and is not the key factor for the AC to make a decision.
2. The experiments are not sufficient, e.g., trade-off between text quality and robustness are not studied.
3. Reviewers have common concerns that many technical details only appear in the appendices. It makes the main body hard to be read.
4. Confusion with previous work such as Christ et al. (2023); Fairoze et al. (2023).

Given most reviewers vote for rejection, AC would follow reviewers' opinion and recommend rejection. AC would encourage the authors to take the above weaknesses into account in the revised version of this paper.

**Additional Comments On Reviewer Discussion:**

Reviewer Bvw7 is the only reviewer with a (slightly) positive rating. However, his/her comments are short and not very informative. Other three reviewers vote consistently for a rejection, two of who are with a score of 3. In the rebuttal phase, the authors only reply to two reviewers, while the other two are ignored with no response. The rebuttal only partially answers the questions raised by the reviewers in the first round of review. For example, it only answers W4 to Reviewer UkDR, while the other four weaknesses are just ignored by the authors. Given these facts, AC would recommend rejection.

---

### Decision · Program_Chairs · 2025-01-22

Reject